***Nat Cell Biol.*** **Author manuscript; available in PMC 2022 June 24.**

# METALIC reveals interorganelle lipid flux in live cells by enzymatic mass tagging

**Arun T. John Peter**[1],

**Carmelina Petrungaro**[2],

**Matthias Peter**[3],

**Benoît Kornmann**[4]

[1]Institute of Biochemistry, Department of Biology, ETH Zurich, Zurich, Switzerland

[2]Institute of Biochemistry, Department of Biology, ETH Zurich, Zurich, Switzerland

[3]Institute of Biochemistry, Department of Biology, ETH Zurich, Zurich, Switzerland

[4]Department of Biochemistry, University of Oxford, United Kingdom

## Abstract

The distinct activities of organelles depend on the proper function of their membranes. Coordinated membrane biogenesis of different organelles necessitates lipid transport from their site of synthesis to their destination. Several factors have been proposed to participate in lipid distribution, but despite its basic importance, *in vivo* evidence linking the absence of putative transport pathways to specific transport defects remains scarce. A reason for this scarcity is the near absence of *in vivo* lipid trafficking assays. Here we introduce a versatile method named METALIC (Mass tagging-Enabled TrAcking of Lipids In Cells) to track interorganelle lipid flux inside cells. In this strategy, two enzymes, one directed to a "donor" and the other to an "acceptor" organelle, add two distinct mass tags to lipids. Mass spectrometry-based detection of lipids bearing the two mass tags is then used to quantify exchange between the two organelles. By applying this approach, we show that the ERMES and Vps13-Mcp1 complexes have transport activity *in vivo*, and unravel their relative contributions to ER-mitochondria lipid exchange.

Correspondence to: Arun T. John Peter; Benoît Kornmann.

Corresponding authors: Arun T. John Peter (arun.johnpeter@unifr.ch), Benoît Kornmann (benoit.kornmann@bioch.ox.ac.uk). Present address: Department of Biology, University of Fribourg, Switzerland

**Author Contributions**
A.T.J.P. and B.K. conceived the study. A.T.J.P. designed and performed all the experiments. C.P. designed and conducted the experiments related to expression of CFAse in mammalian cells. M.P. and B.K. supervised the study. A.T.J.P. and B.K. analyzed data and wrote the manuscript with input from C.P. and M.P.

**Ethical declarations**
Competing interests
The authors declare no competing interests.

## Introduction

Organelle function depends on lipids that constitute their membranes. Membrane lipids not only constitute structural barriers but recruit specific proteins and store energy. Because lipid biosynthesis in eukaryotic cells mostly happens in the endoplasmic reticulum (ER), lipids must be transported to all other cellular membranes. Lipid transport was thought to be a by-product of vesicular trafficking, but the past decade has revealed that cells have evolved non-vesicular mechanisms to mediate the bulk of lipid exchange [1]. Although this transport mode is associated with all organelles, it is especially relevant for organelles like mitochondria that are excluded from vesicular traffic.

Non-vesicular lipid exchange occurs at sites of close contact (10-30 nm) between organelles, where lipid transport proteins (LTPs) solubilize lipids from membranes, shield them from the aqueous milieu in a hydrophobic pocket, and catalyze their exchange between the two membranes. In yeast, the ERMES (ER-Mitochondria Encounter Structure) is a complex of such LTPs implicated in ER-Mitochondria lipid exchange[2–8]. Three subunits, namely Mmm1, Mdm12, Mdm34, harbor a lipid-solubilizing SMP (synaptotagmin-like mitochondrial lipid-binding protein) domain. Surprisingly, however, ERMES deficiency, though resulting in phenotypes including slow growth and defective mitochondrial morphology, does not prevent ER-mitochondria lipid exchange [2,9]. Another LTP implicated in mitochondrial lipid transport is the conserved chorein-N motif-containing protein Vps13, which associates with mitochondria via Mcp1 [10–12]. ERMES inactivation when combined with *vps13* or *mcp1* deletion leads to synthetic lethality [11,13], suggesting that Vps13 partially compensates absence of ERMES. However, although ERMES and Vps13 exhibit lipid transport activity *in vitro* [3–5,14], their redundant role in lipid exchange remains to be proven *in vivo*.

Despite multiple LTPs identified at membrane contact sites, our knowledge on lipid transport and LTP function *in vivo* remains poor, mainly due to limitations in existing tools. Our understanding of phospholipid transport is mainly derived from two methods: a) *in vivo* assays using radiolabeled precursors and, b) *in vitro* lipid exchange assays [14–17]. In typical *in vivo* assays, cells treated with [3]H-serine produce [3]H-phosphatidylserine (PS) via the PS-synthesizing enzyme in the endoplasmic reticulum (ER). As the PS decarboxylase Psd1 produces phosphatidylethanolamine (PE) from PS in the inner mitochondrial membrane (IMM) and the methyltransferases, Cho2 and Opi3, make phosphatidylcholine (PC) from PE exclusively in ER, detection of [3]H-PE and [3]H-PC reflects ER-mitochondria lipid exchange. This assay has limitations. First, it is limited to ER and mitochondria. Second, phospholipases might release labeled lipid headgroups, which can be reincorporated into phospholipids via the Kennedy pathway, independent of interorganelle lipid transport. Finally, a fraction of Psd1 is localized to the ER in addition to the IMM [18] weakening this assay's validity. Although *in vitro* assays monitoring lipid exchange between liposomes [14,17] are useful to test the activity of LTPs implicated in lipid transport, they do not inform about lipid exchange rates, identity, origin and destinations, and regulation of transport routes *in vivo*.

To address these limitations, we have developed an assay called METALIC (Mass tagging-Enabled TrAcking of Lipids In Cells) that exploits enzyme-mediated mass-tagging to measure the exchange of specific lipids between two organelles *in vivo*. Using this approach, we unravel lipid transport activity of Vps13 and ERMES *in vivo* and quantify their relative contributions in ER-mitochondria lipid exchange.

## Results

### Principle of the METALIC assay

In METALIC, a lipid-modifying enzyme is targeted to a "donor" compartment of interest where it chemically modifies lipids, introducing a diagnostic "mass tag." Upon transport to an "acceptor" compartment, mass-tagged lipids encounter a second enzyme that introduces a different "mass tag". The detection of doubly mass-tagged lipids by mass spectrometry thus serves as a proxy to monitor lipid transport between the two compartments.

Importantly, this approach can be combined with metabolic labeling to capture the kinetics of lipid transport. Pulse labeling with deuterated precursors can be used to assess the appearance kinetics of not only the doubly mass-tagged lipids but also the singly-labeled mass tags separately, a proxy for the activity of each enzyme and the metabolic activity of the cell.

### CFAse is active and targetable in yeast

We used cyclopropane-fatty-acyl phospholipid synthase (CFAse), a soluble bacterial enzyme that introduces a methylene group ($-CH_2-$) from *S*-adenosyl methionine (SAM) at double bonds in phospholipid fatty acyl chains (Fig. 1a) [19]. Cyclopropane lipids have similar biophysical properties as their unsaturated counterparts [20], but carry an identifiable +14-Dalton mass tag (Fig. 1a). We expressed CFAse constitutively in yeast and measured cyclopropylation of the most abundant phospholipid species, phosphatidylcholine (PC) by mass spectrometry. The mass spectrum revealed the appearance of peaks 14 Da heavier than the precursor PC species, confirming the enzyme's activity in yeast (Fig. 1b). The modified lipids represented up to 50 % of any species. To test if CFAse can be targeted to specific organelles, we fused targeting sequences (see Methods) to a CFAse-mCherry construct, verified proper localization by microscopy (Fig. 1c) and expression by western blotting (Fig. 2a). Expression of CFAse in organelles did not affect growth, demonstrating that cells tolerated cyclopropane fatty acids (CFA) at the tested organelles (Fig. 1d).

To check if cyclopropane fatty acids were degradable in yeast, we fed cells with CFA (C17:0) and deuterated oleic acid (d-C18:1) (Extended data Fig. 1a). Both fatty acids incorporated in phospholipids (Extended data Fig. 1c). Then we removed the fatty acids from the medium and starved cells from glucose to induce fatty acid break down (Extended Data Fig. 1a). Free C17:0 and d-C18:1 fatty acids had a similar decay profile, comparable to the endogenous oleic acid C18:1 (Extended Data Fig. 1b). This decay was neither due to sequestration of fatty acids in phospholipids, as CFA-PE did not enrich over time (Extended Data Fig. 1c), nor to dilution by newly synthesized lipids, as the biomass remained constant

during glucose starvation (Extended Data Fig. 1d). Taken together, these results suggest the half-life of CFA is similar to their unsaturated precursors.

## CFAse mass-tags various phospholipids in organelles

To assay organelle-targeted CFAse activity, we quantified whole-cell phosphatidylserine (PS), phosphatidylethanolamine (PE), phosphatidylcholine (PC), phosphatidylinositol (PI) and phosphatidylglycerol (PG) in wild-type cells using mass spectrometry. In all cases, we could detect cyclopropylated (+14 Da) lipids (Fig. 2b). Both organelle-specific and distinct patterns were detected. Strikingly, in cells expressing mitochondria matrix-targeted CFAse, CFA-PG and -PE represented the highest fraction relative to other phospholipid species, in line with their precursors being synthesized in the IMM. CFA modification was variable depending on targeting, but not correlated with CFAse expression (Fig. 2a). In particular, CFAse targeted to the ER lumen was less efficient than the one to the cytosolic side of the ER despite being more expressed. This lower activity in the lumen could reflect differences in SAM concentration. Indeed, SAM levels in the endomembrane are unknown. We could exclude the possibility that labeling by ER-lumen-targeted CFAse was due to a minor fraction of untranslocated enzyme as the labeling profile was distinct from untargeted CFAse (Extended Data Fig. 2). In most cases, modified lipids originating from precursors with two double bonds were more abundant than those with a single double bond, consistent with two unsaturated fatty acids having double the chance for CFAse modification. Interestingly, the relative abundance of CFA-PE 32:2, 32:1 and 34:2 was comparable in all compartments except mitochondrial matrix where tagging of mono-unsaturated 32:1 species dominated over the 32:2 and 34:2 species. On the other hand, the relative abundance profile for PC followed the order 32:2>34:2>32:1>34:1 irrespective of the organelle, thus mimicking the abundance observed in the whole cell lipidome of yeast cells (20). Therefore, substrate abundance can explain differences in modification efficiency. Taken together, these results demonstrate that the bacterial CFAse can specifically and efficiently tag phospholipids in organelles.

## Strategy to monitor ER-mitochondria lipid exchange *in vivo*

To monitor ER-mitochondria lipid transport, we utilized the exclusive ER localization of the PE methyltransferases Cho2 and Opi3 to introduce one of the two mass tags. We targeted CFAse to the mitochondrial matrix to introduce the other mass tag. Since both CFAse and the methyltransferases use SAM as a methylene or methyl donor, respectively, we pulse-labeled cells with deuterated methionine (d-methionine) [21] and monitored the appearance of both singly- and doubly-labeled phospholipids, the latter being indicative of lipid transport between the ER and mitochondria (Fig. 3a,b,c).

As for the singly-labeled species, we monitored the +9 Da PC species resulting from the triple methylation of the headgroup at the ER (three deuterated -$CH_3$ groups being 9 Da heavier than three non-deuterated ones), and the +16 Da PC species resulting from cyclopropylation of PC at mitochondria. The doubly labeled species has a +25 Da (9+16) mass shift, (Fig. 3b). which can either result from the head group-labeled PC transported to mitochondria (Fig. 3a, Route 1) or cyclopropane-labeled PE transported to the ER (Fig. 3a, Route 2). Our measurements thus assess both transport directions.

Upon pulse labeling in wild-type cells, we observed a time-dependent increase in the fraction of deuterated headgroups and deuterated cyclopropanes, among the most abundant PC species (i.e., 32:2, 32:1 and 34:2, Fig. 3c, red line). While incorporation of deuterated headgroups saturated close to 100%, deuterated cyclopropanes in PCs saturated at lower values, consistent with the fact that CFAse only modifies a fraction of lipids (Fig. 2). The appearance kinetics of deuterated headgroups and deuterated CFAs was consistent with a model where lipid synthesis accommodates the requirement for biomass increase (as assessed by the change in $OD_{600}$, Extended data Fig. 3A-D). The $t_{3.5}$ timepoint was usually the most discrepant, likely stemming from the fact that the deuterated methionine labeling at $t_0$ involves a sudden tenfold increase in methionine availability to which cells might need to adapt.

For all major PC species, the fraction of +25 Da double-labeled lipids increased over time (Fig. 3c, red line), indicative of ER-mitochondria lipid transport, the kinetics of which was again consistent with expectations (*i.e.,* corresponding to the product of deuterated headgroup and cyclopropane fractions, Extended Data Fig. 3E).

To validate the specificity of this strategy, we tested its dependency on Sam5, the major transporter of SAM across the IMM[22]. Since CFAse activity is dependent on SAM in the mitochondrial matrix, our prediction was that in the *sam5* mutant, mass-labeling should be severely impaired. Indeed, while headgroup labeling at ER was similar to wild-type (the small difference might be accounted for by a slower growth rate of *sam5* mutants), the incorporation of deuterated cyclopropane, as well as double mass-labeling were severely reduced in *sam5* mutant cells (Fig. 3c, grey line). While some incorporation was observed in early timepoints, consistent with the notion that early incorporation data are perturbed by cellular adaptation to high methionine, cyclopropane incorporation returned to background levels at later timepoints, confirming near complete CFAse targeting to the mitochondrial matrix with little activity outside it. Taken together, these results highlight the robustness and sensitivity of METALIC for monitoring ER-mitochondria phospholipid exchange *in vivo*.

### An auxin-inducible degron system to inactivate ERMES

To assess the redundant roles of ERMES and Vps13-Mcp1 complexes in lipid exchange, we sought to assay lipid transport upon inactivation of both pathways. As co-deletion is synthetically lethal, we built an inducible system to acutely inactivate ERMES. We C-terminally fused Mdm12 to an auxin-inducible degron (AID) [23] and expressed *At*TIR1, a plant auxin-dependent adapter for E3 ubiquitin ligases. 1 hour of auxin treatment efficiently depleted Mdm12-AID (Fig. 4a), causing typical morphological changes in mitochondria (Fig. 4b), which was unhindered by the loss of Vps13 or Mcp1 (Fig. 4d). Finally, in the presence of auxin, cells expressing Mdm12-AID grew slower, and this was exacerbated by concomitant deletion of *VPS13* or *MCP1* (Fig. 4c) as expected[11,13,24]. These results confirm that auxin-dependent Mdm12 depletion rapidly inactivates ERMES, thus allowing to test its lipid transport activity *in vivo* using the METALIC assay.

## Both ERMES and Vps13 contribute to phospholipid exchange

To unravel the roles of ERMES and Vps13-Mcp1 complexes in ER-mitochondria lipid exchange, we pulse-labeled cells with d-methionine and assayed mass tag-labeling upon inactivation of either one or both pathways (Fig. 5a). Headgroup labeling kinetics was similar in all the LTP mutants, indicating that despite different growth phenotypes, cells are metabolically active and generate new PC headgroups at comparable rates (Fig. 5b, c, top panels).

To address the contribution of the Mcp1/Vps13 pathway alone to lipid transport, we assayed mass tag-labeling in *MDM12-AID mcp1* cells, without auxin (- auxin) to maintain ERMES function (Fig. 5b). First, we quantified all species with deuterated headgroup or cyclopropane rings with the exception of doubly mass-labeled species, to assess label incorporation independent of transport. While we observed a similar increase in headgroup labeling, increased modification of the cyclopropane ring was transient, probably reflecting faster labeling at the headgroup than at the cyclopropane ring, and thus by the end of the experiment most PC molecules bore deuterated headgroups. As shown in Fig. 5b, *MCP1* deletion alone impacted neither headgroup nor cyclopropane labeling. We then quantified doubly mass-labeled lipids (+25 Da), indicative of ER-mitochondria lipid exchange. In this set-up, doubly mass-labeled species increased monotonously in *MDM-12-AID* (surrogate wild-type) and *MDM-12-AID mcp1* cells. While PC 32:2 was not affected, after 14.5 hours, there was a mild (25%) but significant reduction in double mass labeling of PC 32:1 (*p*-value = 0.038) and a non-statistically significant reduction in PC 34:2 (*p*-value = 0.32) (Fig. 5b, lower panels).

To assess the role of ERMES, we treated cells bearing *MDM12-AID* with auxin for 7 hours before pulse-labeling (Fig. 5a, Extended Data Fig. 4). While headgroup and cyclopropane labeling was unaffected, double mass-labeling (+25 Da species) was reduced (Fig. 5c, red lines), especially for the PC 32:2 species.

Finally, we assessed mass tagging in *MDM12-AID* cells with either *VPS13* or *MCP1* deleted. We observed a slight reduction in headgroup and cyclopropane labeling (at least for the PC32:1 species, Fig. 5c, blue and green lines). Strikingly, however, double mass labeling was reduced close to background levels by co-inactivation of ERMES and either Vps13 or Mcp1, particularly for PC 32:1.

Thus, while both pathways might show some specificity with regard to the transported phospholipids, these results demonstrate that ERMES and Vps13-Mcp1 complexes function in ER-mitochondria lipid exchange *in vivo*, providing a biochemical basis for their genetic redundancy.

## CFAse mass-tags phospholipids in mammalian cells

To assess if METALIC could be used in higher eukaryotes, we expressed various mCherry-tagged organelle-targeted CFAse constructs in HeLa cells under the control of a doxycycline-inducible promoter by lentiviral transduction. Immunofluorescence and western blotting confirmed proper localization and expression (Fig. 6a,b, Extended Data Fig. 5a). LC-MS showed that lipids with mass consistent with cyclopropylation could

be detected even in cells expressing no CFAse. This is likely due to the presence of ether-linked plasmalogen lipids, containing fatty alcohols that are 14 Da lighter than their fatty acid counterpart with same carbon number. Thus, a C16 fatty acid with cyclopropane modification (C17) has a molecular weight identical to a C18 fatty alcohol. Nevertheless, we could observe an increase in abundance of species corresponding to cyclopropane lipids for most constructs and most lipids (Fig. 6c). The changes observed were consistent with the enzyme's subcellular localization. For instance, CFAse targeted to either the mitochondrial matrix or intermembrane space was most efficient at modifying PG, a lipid virtually exclusively found in mitochondrial membranes. By contrast, the same constructs were very inefficient at modifying PS, a lipid which is rare in mitochondrial membranes. Importantly, expression of the enzyme and lipid modification did not affect cell survival as assessed by Trypan Blue staining (Extended Data Fig. 5b). Together these data indicate that CFA synthase can be utilized to modify lipids in an organelle-specific way in higher eukaryotes.

## Discussion

Here we demonstrate the utility of METALIC, a versatile strategy to probe interorganelle lipid transport *in vivo* using ER-mitochondria as a model organelle pair. Our survey ties hampering loose ends in our understanding of ER-mitochondria lipid exchange. It demonstrates the contribution of two candidate pathways, for which direct *in vivo* evidence had been thus far missing or incomplete [2–5,8,25]. We observe that Vps13-Mcp1 pathway contributes minimally to ER-mitochondria lipid exchange, correlating with the observation that neither *VPS13* nor *MCP1* deletion affects mitochondria morphology or yeast growth. On the other hand, contribution of ERMES to lipid transport is substantial, in line with the strong mitochondrial and growth phenotypes of *ermes* mutants. Moreover, the two pathways function in a redundant fashion, accounting for the bulk of lipid transport between the two organelles, potentially explaining the synthetic lethality of mutants lacking them.

Interestingly, the lipid transport defect in *ermes* mutants is particularly striking for doubly unsaturated species 32:2 and 34:2. By contrast, the 32:1 species was most affected in *ermes vps13* or *ermes mcp1* double mutant cells. In fact, transport of this lipid was modestly but significantly affected by the loss of *MCP1* alone. Together, these findings suggest that lipid-binding pockets of LTPs could have preferences for specific fatty acids.

The fact that CFAse can be directed to multiple compartments makes it possible to study phospholipid transport between ER and any organelle of interest, as long as targeting is stringent. In the case of mitochondrial matrix, both microscopy and the analysis of the *sam5* mutant indicates that mistargeting of CFAse is negligible. In fact, we do not know if the residual activity of matrix-directed CFAse in the *sam5* mutant is due to partial enzyme mistargeting or residual mitochondrial membrane permeability to SAM in the absence of Sam5.

Our data show that cyclopropane lipids can be synthesized and transported in yeast and human cells without perturbing their function. The approach is complicated in mammalian cells by plasmalogens with the same m/z as cyclopropane-modified lipids. This limitation could be overcome by using tandem mass-spectrometry which could detect different

fragmentation product for plasmalogen and cyclopropane lipids, or other LC approaches to discriminate the two types. Of note, *C. elegans* worms feeding on bacteria incorporate CFA into their lipidome[26,27], indicating that CFAse can likely be used in invertebrate systems. Therefore, even if their behavior is not identical to unsaturated lipids, cyclopropane lipids can serve as useful tools within the METALIC approach to assay lipid transport, and unravel relative differences in different genetic backgrounds (lipid transport mutants) or physiological conditions. In addition to MS-based methods, CFA-lipids can be detected by Raman spectroscopy, potentially allowing single-cell resolution[28].

One limitation of the enzymes chosen here is their requirement for SAM. Most of the known SAM-requiring enzymes in yeast have active sites in the cytoplasm, nucleus or mitochondria[29], suggesting that SAM might not be available in the lumen of other organelles. Our results indicate that ER-lumen targeted CFAse can modify lipids in this compartment (Fig. 2). In addition, SAM levels in the lumen of endocytic compartments can likely be manipulated by adding SAM to the culture medium, which can be endocytosed by bulk flow. Nevertheless, to overcome the issue of SAM availability, we chose to target most CFAse constructs to organelles' cytosolic face (Fig. 1c). Whether enzymes tethered to a membrane's cytosolic side might act on other membranes *in trans* at interorganelle contacts needs to be verified for chosen organelle pairs. One last limitation of the approach is that it does not inform on transport directionality (see Fig. 3a, route 1 vs. route 2), nor whether it is direct or involves intermediate compartments, as is likely the case for the Mcp1/Vps13 pathway (see below). Therefore, any rate calculated with METALIC cannot be used as a direct measure of lipid exchange. However, the effect of perturbations on lipid traffic can be measured with METALIC, as we show here for ER-mitochondria lipid transport. One obvious caveat of enzyme-based methods is that any perturbation can affect either lipid transport or mass tag incorporation. In METALIC, the incorporation rates by both enzymes can be surveyed independently and used to normalize the rate of appearance of the doubly mass-tagged (and therefore transported) species.

The involvement of multiple redundant LTPs in interorganelle lipid transport appears to be the rule rather than the exception. Indeed, among ~40 putative LTPs identified in yeast, none is truly essential for growth, indicating redundant mechanisms at play. Here, we study two pathways allowing exchange of lipids between mitochondria and the endomembrane system. While the ERMES complex localizes to ER-mitochondria contacts and therefore, likely catalyzes direct lipid exchange between the two compartments, the Vps13/Mcp1 has been found at mitochondria-vacuoles[11,13] and mitochondria-endosome contacts[24]. The various localization of these complexes indicates that lipids can use alternate routes that may or may not involve intermediate organelles, to transit from their synthesis site to their destination. The direct vs. indirect nature of the transport pathways, in addition to the intrinsic preferences of different LTPs, might explain potential lipid specificities observed here. Deciphering the contribution of these many LTPs, their redundancy, and their preferences, therefore, constitutes an important challenge in cell biology. However, despite the central contribution of lipids to many cellular functions, our knowledge significantly lags behind DNA, RNA and proteins, as we do not have the equivalent tools (PCR, GFP-tagging). The development of METALIC thus takes an important step forward and paves the way to elucidate LTP function and lipid transport processes *in vivo*.

## Methods

### Yeast strains and plasmids

Yeast strains, plasmids and primers used in this study are listed in Supplementary Table 1. Yeast cells were cultured at 30 °C in synthetic defined (SD) medium with 2% glucose (2% glucose, 0.5% NH$_4$-sulfate, 0.17% yeast nitrogen base and amino acids). Genomic integration of PCR fragments was done by homologous recombination [30,31]. Gene deletions were confirmed by colony PCR. Growth curves were obtained using Clariostar equipped with the manufacturer's software (version 5.4). CFAse ORF was amplified from *E. coli*. A CFAse-mCherry construct was targeted to different organelles using the following targeting sequences; for the ER membrane, the C-terminal 20 residues of Ubc6 (230-250); for the ER lumen, the signal sequence of Kar2 (aa 1-41) at the N-terminus and a HDEL signal at the C-terminus; for the outer mitochondrial membrane, aa 1-30 of Tom70; for the mitochondrial matrix, aa 1-69 of subunit 9 of the F$_0$-ATPase from *N.crassa*; for the peroxisome, full-length Pex3 at the N-terminus; for the plasma membrane, the PH domain of Osh1 (aa 268-388); for the vacuole, full-length Vac8 at the N-terminus.

### Mammalian cell culture

HeLa cells were cultured in MEMα-medium, supplemented with 10 % fetal calf serum (FCS) and 1 % penicillin and streptomycin (P/S). They were incubated at 37 °C and 5 % CO$_2$. Stable cell lines were generated by lentiviral transduction as published before[32]. CFAse-mCherry construct was targeted to different organelles using the following targeting sequences: For lysosomal targeting the 407 aa of Lamp1 were fused N-terminally to CFAse; for peroxisomal targeting a SKL signal at the C-terminus; for mitochondrial intermembrane space targeting the presequence of Smac1 (aa 1-59) at the N-terminus; for mitochondrial matrix targeting the presequence of Cox8 (aa 1-35) at the N-terminus.

### Microscopy

Cells were grown to mid-log phase in synthetic defined (SD)–uracil medium for selection of the mitochondrial matrix-targeted CFAse-mcherry plasmid. Mammalian cells were incubated in MEMα-medium containing 10 % FCS and 1 % P/S. The expression of the CFAse-mCherry constructs was induced with 1 mM Doxycyline overnight. Prior to the imaging, the medium was exchanged wit PBS.

Images were acquired using a DeltaVision MPX microscope (Applied Precision) equipped with a 100× 1.40 NA oil UplanS-Apo objective lens (Olympus), a multicolor illumination light source, and a CoolSNAPHQ2 camera (Roper Scientific). Image acquisition was done at room temperature. Images were deconvolved with Deltavision SoftWoRx software (version 6.5.2) using the manufacturer's parameters. Images were processed further using FIJI ImageJ (version 1.53c) bundle.

### Protein extraction and western blotting

For yeast,1 OD$_{600}$ of mid-log phase cells were collected by centrifugation and precipitated using 10% trichloroacetic acid for 20 min at 4°C. After centrifugation at 13,000 *g* for 5 min, pellets were washed with ice-cold acetone. Pellets were air-dried and resuspended in 30 μl of

1× SDS sample buffer (60 mM Tris, pH 6.8, 2% SDS, 10% glycerol, 5% 2-mercaptoethanol, and 0.005% bromophenol blue), and boiled for 3 min. For mammalian cells, $10^6$ cells were scraped off in 100 μl of SDS sample buffer and heated at 96 °C for 10 min. Samples were resolved on a 12% SDS-PAGE gel, and after transfer on a PVDF membrane, proteins were detected using specific antibodies. The following antibodies were used: mouse anti-Pgk1 antibody (Invitrogen, 459250, 1:3000 dilution), rat anti-RFP antibody (Chromotek, 5F8, 1:1000 dilution), mouse anti-FLAG antibody (Sigma, F1804, 1:1000), rabbit anti-mCherry antibody, ab167453, 1:1000) and horseradish peroxidase-coupled secondary antibody (Bio-Rad, 170-6516; 1:10,000 dilution), Western blots were imaged using the Fusion FX system (Vilber) equipped with the FusionCapt Advance FX7 software (version 17.03).

## Pulse-labeling, lipid extraction and mass spectrometry analysis

Pre-cultures in SD medium were diluted to 0.8 $OD_{600}$/ml in 25 ml and treated with 0.5 mM auxin for 7 hours. Next, cells were pulse labeled with 2 mM deuterated methionine and grown at 30 °C. At the indicated time points, 8 $OD_{600}$ of cells was pelleted, snap-frozen and stored at -80°C. Lipids were extracted as described previously with minor modifications [33]. Briefly, cells were washed in ice-cold water and subsequently resuspended in 1.5 ml of extraction solvent containing ethanol, water, diethyl ether, pyridine, and 4.2 N ammonium hydroxide (v/v 15:15:5:1:0.18). After the addition of 300μL glass beads, samples were vortexed vigorously for 5 minutes and incubated at 60 °C for 20 min. Cell debris were pelleted by centrifugation at 1,800 ×g for 10 min and the supernatant was dried under a stream of nitrogen. The dried extract was resuspended in 1 ml of water-saturated butanol and sonicated for 5 min in a water bath sonicator. 500 μl of water was added and vortexed further for 2 min. After centrifugation at 3000 ×g, the upper butanol phase was collected, dried under a stream of nitrogen and resuspended in 50% methanol for lipidomics analysis.

LC analysis was performed as described previously with several modifications [34]. Phospholipids were separated on a nanoAcquity UPLC (Waters) equipped with a HSS T3 capillary column (150 m x30mm, 1.8 m particle size, Waters), applying a 10 min linear gradient of buffer A (5 mM ammonium acetate in acetonitrile/water 60:40) and B (5 mM ammonium acetate in isopropanol/acetonitrile 90:10) from 10% B to 100% B. Conditions were kept at 100% B for the next 7 min, followed by a 8 min re-equilibration to 10% B. The injection volume was 1 μL. The flow rate was constant at 2.5 μl/min.

The UPLC was coupled to QExactive mass spectrometer (Thermo) by a nanoESI source (New Objective Digital PicoView® 550) equipped with the Thermo QExactive XCalibur software (version 4.0.27.10). The source was operated with a spray voltage of 2.9 kV in positive mode and 2.5 kV in negative mode. Sheath gas flow rate was set to 25 and 20 for positive and negative mode, respectively. MS data was acquired using either positive or negative polarization, alternating between full MS and all ion fragmentation (AIF) scans. Full scan MS spectra were acquired in profile mode from 107-1600 m/z with an automatic gain control target of 1e6, an Orbitrap resolution of 70,000, and a maximum injection time of 200 ms. AIF spectra were acquired from 107-1600 m/z with an automatic gain control value of 5e4, a resolution of 17,500, a maximum injection time of 50 ms and fragmented with a normalized collision energy of 20, 30 and 40 (arbitrary units). Generated fragment

ions were scanned in the linear trap. Positive-ion-mode was employed for monitoring PC and negative-ion-mode was used for monitoring PS and PE. Lipid species were identified based on their m/z and elution time. We used a standard mixture comprising PS 10:0/10:0, PE 17:0/17:0, PC 17:0/17:0, PG 17:0/17:0 and PI 12:0/13:0 for deriving an estimate of specific elution times. Lipid intensities were quantified using the Skyline (version 21.2.0.369) software [35]. For each phospholipid, signal was integrated for the precursor species $(m)$, cyclopropane species $(m_{+14})$ and species that appear upon pulse-labeling with deuterated methionine $(m_{+9}, m_{+16}, m_{+23}, m_{+25})$. Fraction of cyclopropylated species (Fig. 2) upon constitutive expression of CFAse was calculated as $(m_{+14}) / (m + m_{+14})$. Fraction of labeled headgroups (Fig. 3), was calculated as $(m_{+9} + m_{+23} + m_{+25}) / (m + m_{+9} + m_{+14} + m_{+16} + m_{+23} + m_{+25})$. Fraction of labeled cyclopropane (Fig. 3) was calculated as $(m_{+16} + m_{+25}) / (m + m_{+9} + m_{+14} + m_{+16} + m_{+23} + m_{+25})$. Fraction of labeled headgroups and cyclopropane, independent of transport (Fig. 5), were calculated as $(m_{+9} + m_{+23}) / (m + m_{+9} + m_{+14} + m_{+16} + m_{+23} + m_{+25})$ and $(m_{+16}) / (m_{+14} + m_{+16} + m_{+23} + m_{+25})$, respectively. Fraction of doubly labeled mass-tagged species (Fig. 3 and 5) was calculated as $(m_{+25}) / (m + m_{+9} + m_{+14} + m_{+16} + m_{+23} + m_{+25})$. A step-by-step protocol describing the METALIC approach can be found at Protocol Exchange[36].

### Lipid extraction and analysis for mammalian cells

$10^6$ cells were scraped off, pelleted and resuspended in 125 µl water. They were transferred to glass tubes and 250 µl cold extraction solvent (Methanol: 0.1 N HCl (1:1)) was added. The suspension was vortexed for 1 min before 250 µl cold Chloroform was added. After 15 min incubation, solution was spun for 20 min at 3500 g at 4 °C. The lower phase was transferred into a fresh glass tube and dried under a stream of nitrogen. For MS analysis, samples were resuspended in 100 µl Chloroform : Methanol : deionized water (73:23:3 v/v/v) to a concentration of 2 ng/µl.

For MS analysis, lipids were separated on a Diol-column (MultoHigh 100 Diol 5µ Hilic Column (CS-Chromatographie Service GmbH)) applying a 15 min linear gradient of mobile phase A (80 % Chloroform, 19.5 % Methanol, 0.5 % Ammonium hydroxide) and B (60.3 % Chloroform, 34.2 % Methanol, 5 % deionized water, 0.5 % Ammonium hydroxide) from 0% B to 100% B. Conditions were kept at 100% B for the next 11 min, followed by a 5 min re-equilibration to 0% B. The injection volume was 2 µL (4 µg lipids). MS was performed with a Advion ExpressIon L. Scan mode: 400-1600 m/z; Total scan time: 50 min; Scan speed: 2500; Scan time: 240 ms.

### Statistics and reproducibility

For Fig. 1, 2a, 4, 6a-b and Extended Data Fig. 5a, the presented data are representative results from at least three independent experiments, unless otherwise specified in the figure legends. For Fig. 2b, 3, 5 and Extended Data Fig. 1, 2, 3, 4, 5b, quantifications were derived from three independent experiments or clones, unless specified otherwise in the figure legends. Whenever possible, individual data points of individual experiments are shown. GraphPad Prism 8, Windows Excel (version 2108) and Rstudio (1.4.1103) was used to analyse and plot data. No statistical method was used to predetermine sample size. No

data were excluded from analyses. The investigators were not blinded to allocation during the experiments and outcome assessment.

## Extended Data

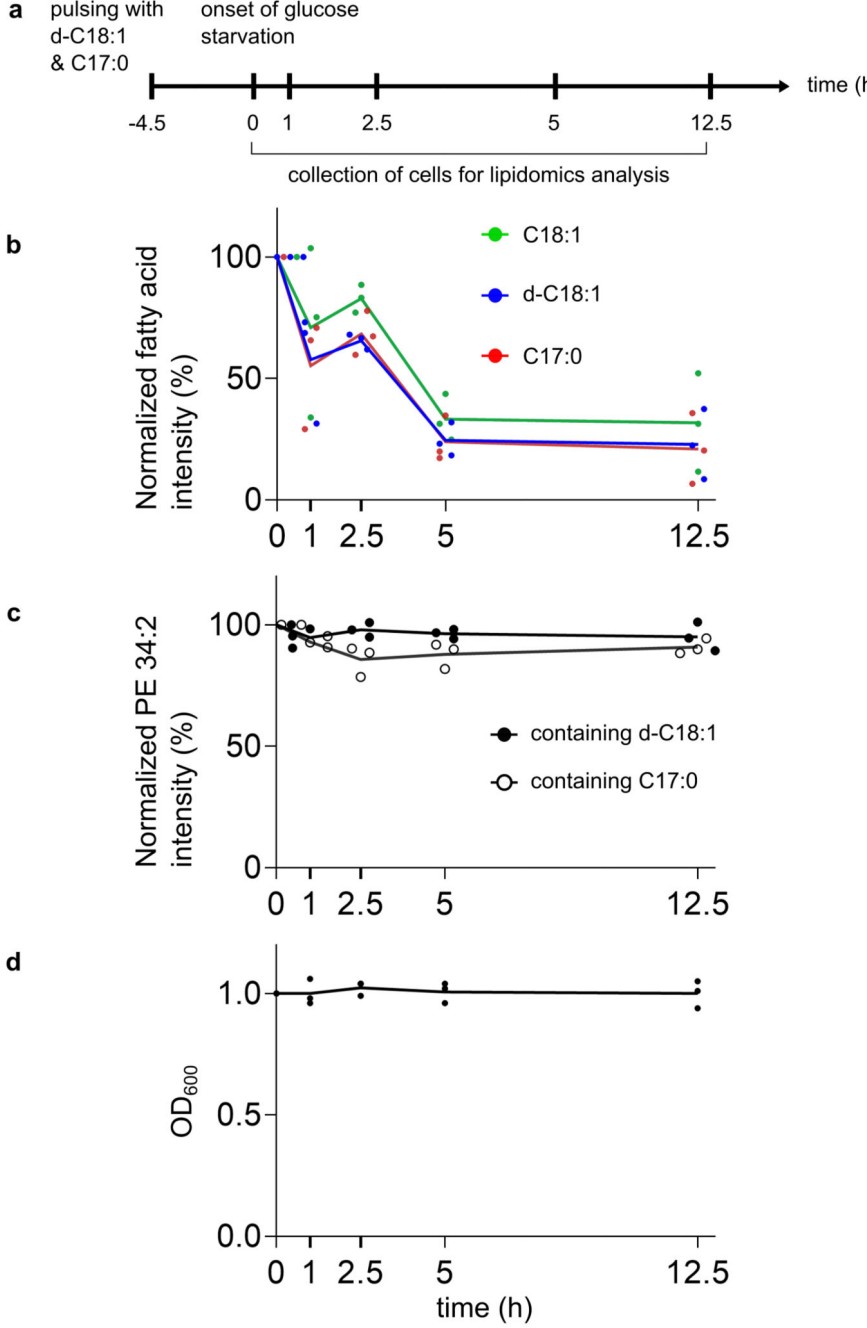

**Extended Data Fig. 1. Cyclopropane fatty acid and its unsaturated counterpart are metabolized in a similar manner**

**a**, Depiction of the time points at which wild-type cells were collected for lipidomics analysis upon onset of glucose starvation, post pulse-labeling with deuterated oleic acid

d-C18:1 and cyclopropane fatty acid C17:0 for 4.5 hours. **b**, Line plot showing the intensity of the endogenous oleic acid (C18:1) and the ectopically provided deuterated oleic acid (d-C18:1) or cyclopropane fatty acid (C17:0) over time upon onset of glucose starvation. **c**, Line plot showing the intensity of a phosphatidylethanolamine (PE) species that has either incorporated d-C18:1 or C17:0 as one of its fatty acid moieties. **d**, The growth profile of pulse-labeled wild-type cells post glucose starvation. Source numerical data are available in source data.

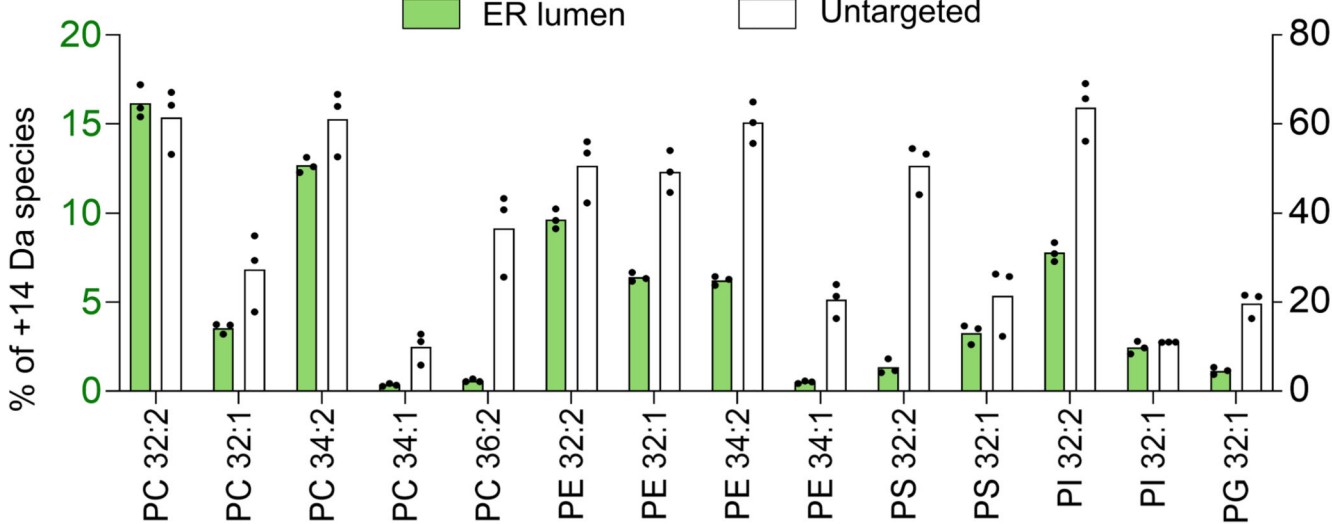

**Extended Data Fig. 2. CFAse targeted to the ER lumen has the different modification profile than its untargeted counterpart.**

Bar plot showing the percentage of each indicated phospholipid species that is mass-tagged upon constitutive expression of organelle-targeted CFAse in yeast. Values for the cytosolic (untargeted) CFAse are read on the right axis, values for the ER-lumen targeted CFAse are read on the left axis. Percentage values represent the mean derived from experiments done on three independent clones. Source numerical data are available in source data.

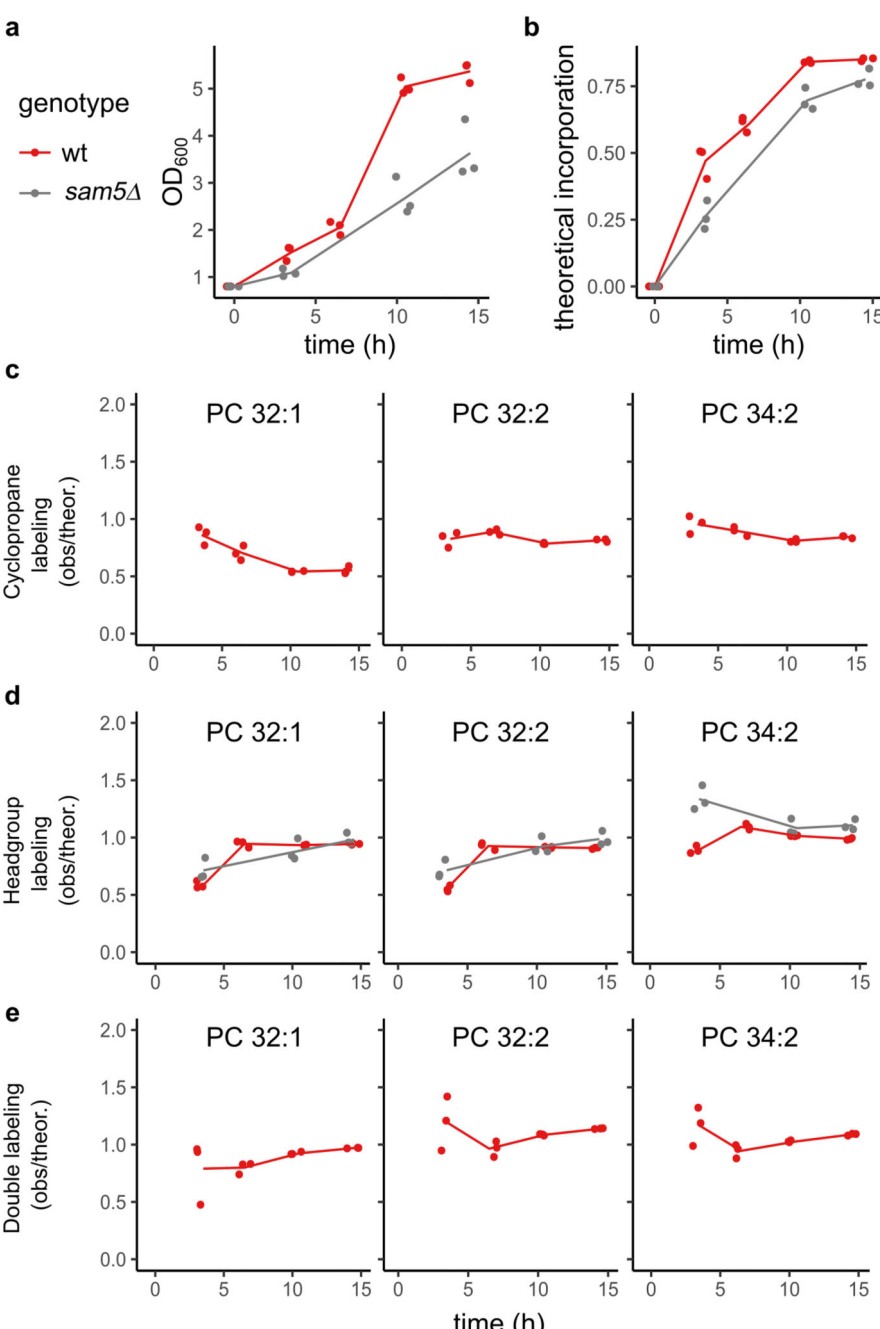

**Extended Data Fig. 3. Theoretical deuterium incorporation.**

**a,** $OD_{600}$ of the cultures used in Figure 3 (Note that OD measurements are missing for the *sam5Δ* strains at $t_{6.5}$). **b,** Theoretical ratio of newly synthesized over total lipid expected if the abundance of newly synthesized lipids is proportional to the change of $OD_{600}$ in the culture minus the starting $OD_{600}$, *i.e:* Theoretical incorporation = $(OD_{600}\text{-}0.8)/OD_{600}$. **c,** Ratio of the observed incorporation to the theoretical incorporation (as displayed in B) for CFAse activity, where CFA labeling is the sum of the +16 and +25 kDa species. **d,** Ratio of the observed incorporation (as displayed in Fig. 3C) to the theoretical incorporation

(as displayed in B) for methyltransferase activity. **e**, Ratio of the observed double label incorporation (as displayed in Figure 3) to the theoretical one calculated as the product of the headgroup and CFA labeling. Source numerical data are available in source data.

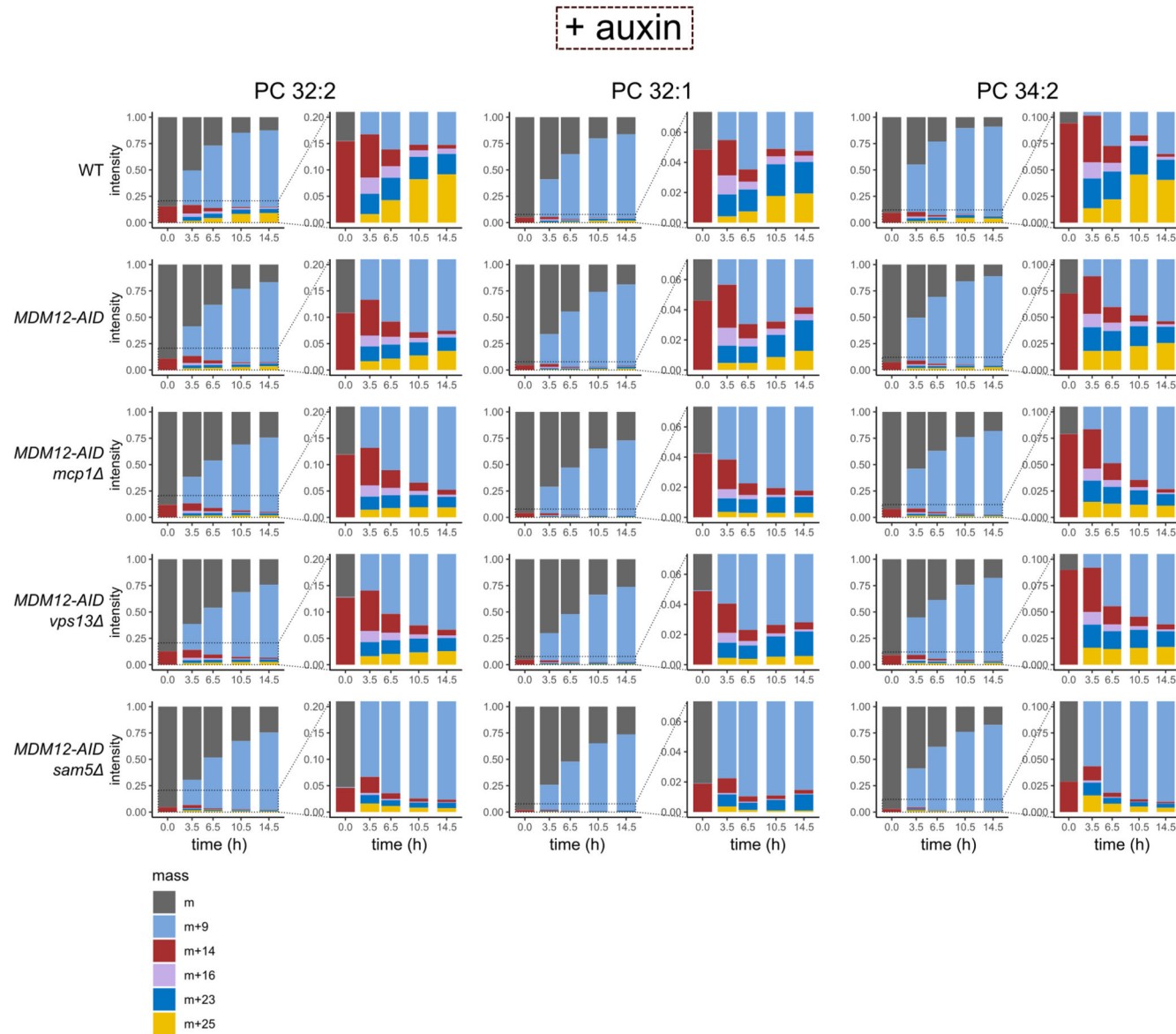

**Extended Data Fig. 4. Raw proportion of each lipid species.**
Normalized bar plot showing the relative changes in intensities of different labeled species with time (hours), from the onset of d-methionine labeling in auxin-treated cells. The changes are shown for three different phospholipid species, PC 32:2 (**a**), PC 32:1 (**b**) and PC 34:2 (**c**). Inset shows magnified views to better follow the intensity changes in species with a cyclopropane ring. Source numerical data are available in source data.

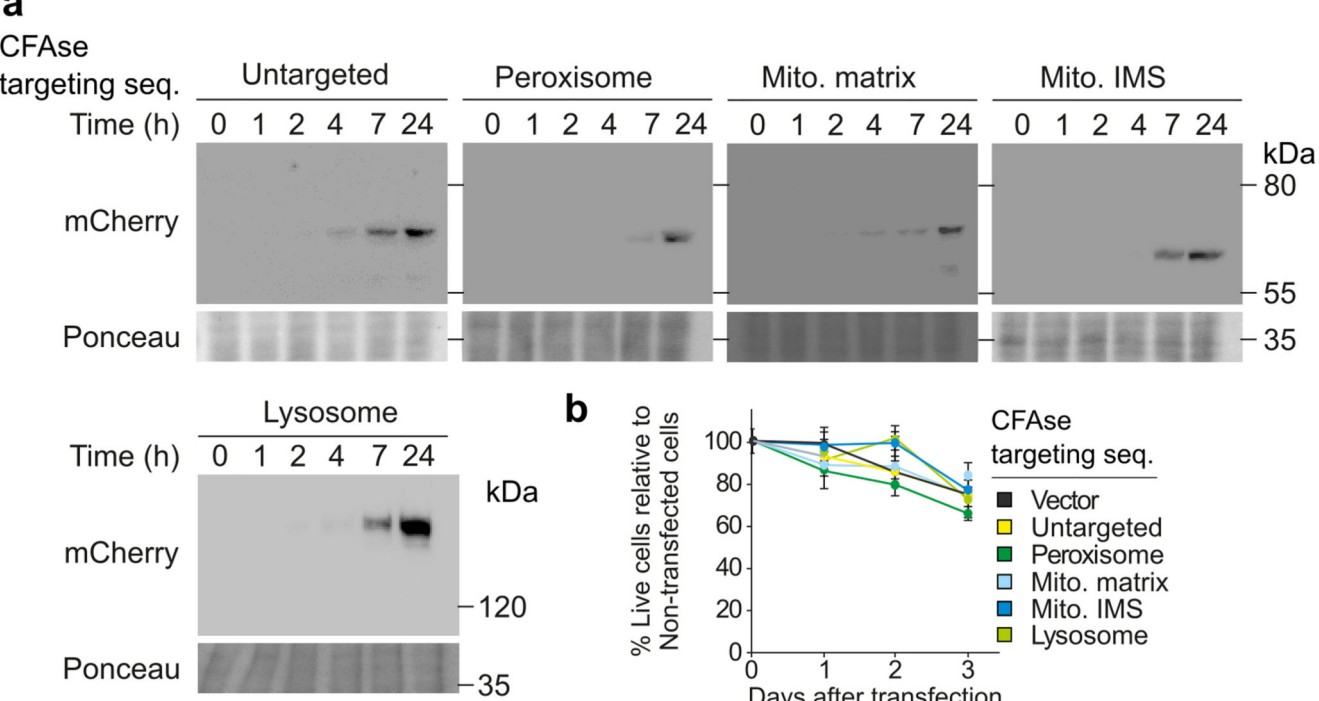

**Extended Data Fig. 5. CFAse can be expressed inducibly and is not toxic in HeLa cells.**
**a,** Expression of mCherry-tagged CFAse variants in HeLa cells after doxycycline induction (1 µg/ml) for one hour. Expression was monitored by western blot at the indicated time points. This experiment was performed once but similar results were obtained in Fig. 6. **b,** Cell viability upon CFAse expression. CFAse expression was induced by doxycycline (1 µg/ml) for the indicated time. The medium was replaced every 24 hours; Data represent the mean +/- S.D from three independent experiments. Source numerical data and unprocessed blots are available in source data.

## Supplementary Material

Refer to Web version on PubMed Central for supplementary material.

## Acknowledgements

We are thankful to members of the Peter and Kornmann laboratories for insightful discussions and helpful suggestions. Microscopy analysis was carried out at the ETH Zürich ScopeM facility, and we thank Dr. Tobias Schwarz for outstanding technical support. Lipidomics measurements were performed at the Functional Genomics Center Zurich (FGCZ). We especially thank Dr. Sebastian Streb and Dr. Endre Laczko of the FGCZ Metabolomics division for establishing and optimizing lipidomics workflows, and for excellent technical guidance. Ylp204-pADH1-*At*TIR1-9myc was a gift from Helle Ulrich (Addgene plasmid #99532), and we are grateful to Dr. Philipp Kimmig for providing us the *At*TIR1 construct in a pRS303 backbone.

The Kornmann lab is supported by grants from the Swiss National Science Foundation (SNSF, 31003A_179549) and the Wellcome Trust (214291/A/18/Z). Work in the Peter laboratory was supported by the SNSF (310030_200426 / 1), the Synapsis Foundation (2020-PI03) and ETH Zürich. A.T.J.P was supported by ETH Zürich/Institute of Biochemistry, Spark grant of the SNSF (CRSK-3_190364) and a FreeNovation grant of the Novartis Foundation (TE-70768 FreeNovation), C.P. by an EMBO fellowship (EMBO ALTF 298-2016).

## Data availability

"Mass spectrometry data have been deposited to the MetaboLights metabolomics repository (dataset identifier MTBLS3415). Numerical source data (with all independent repeats) and unprocessed images of gels and blots are provided in the Source Data files. All other data supporting the findings of this study are available from the corresponding author on reasonable request." Source data are provided with this paper.

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

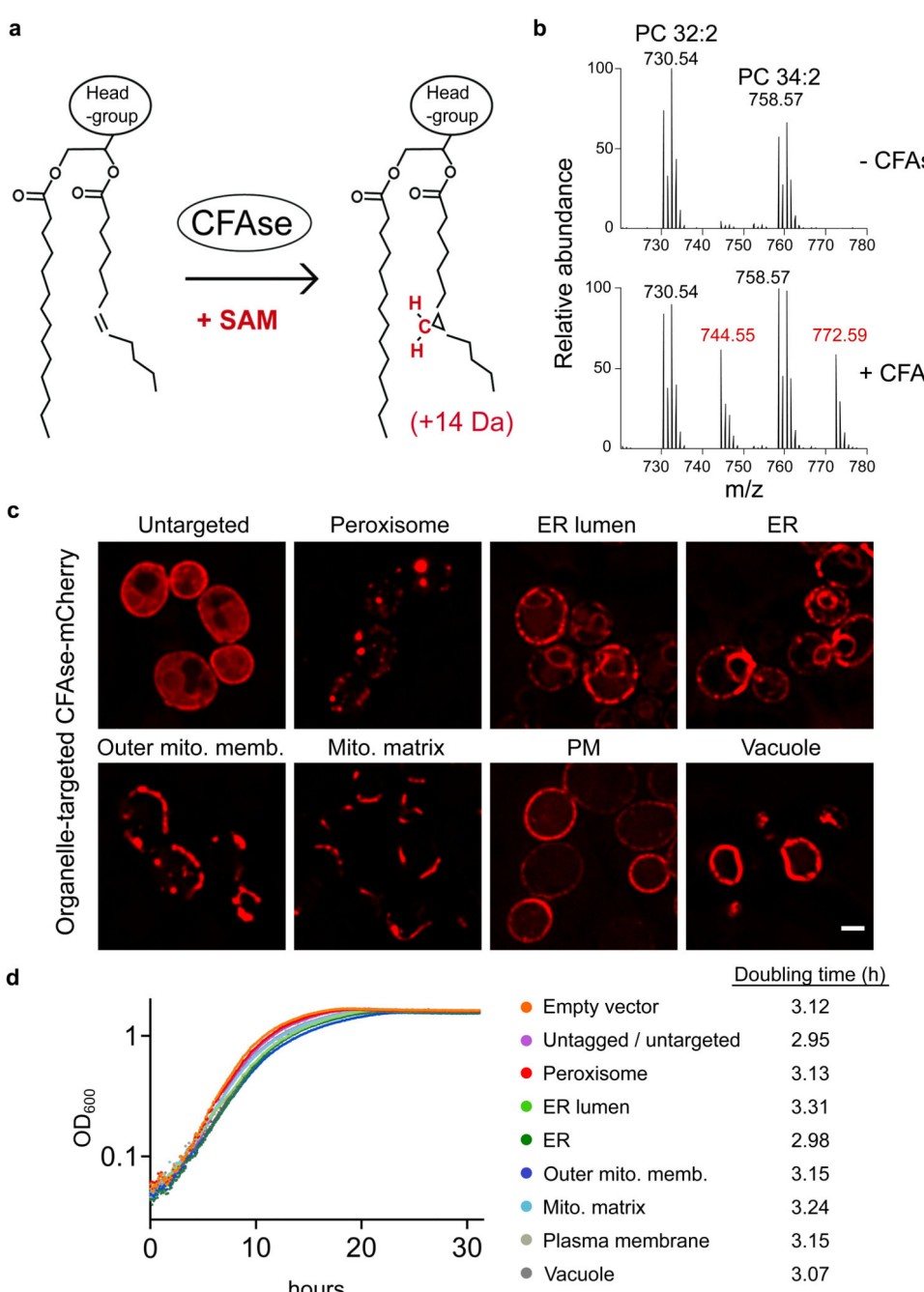

**Figure 1. CFAse is active and targetable in yeast.**

**a**, Scheme depicting the CFAse reaction. CFAse adds a methylene group to double bonds on fatty acyl chains in phospholipids irrespective of their head group, using *S*-adenosyl methionine (SAM) as a co-factor, resulting in a +14 Dalton mass-shift. **b**, CFAse is active in yeast. Mass spectrum showing phosphatidylcholine species PC 32:2 (730.54 m/z) and PC 34:2 (758.57 m/z) and their mass-tagging upon expression of CFAse in yeast. **c**, Localization of mCherry-tagged CFAse to different organelles by fusion to targeting sequences (see Material and Methods). Scale bar, 2 μm. Similar results were obtained in three independent

experiments **d**, Semi-log plot of the growth of yeast cells expressing mCherry-tagged CFAse constructs targeted to different organelles was monitored by $OD_{600}$ measurements. Doubling time for each construct is indicated in hours. For each construct, growth was monitored for three independent clones. Source numerical data are available in source data.

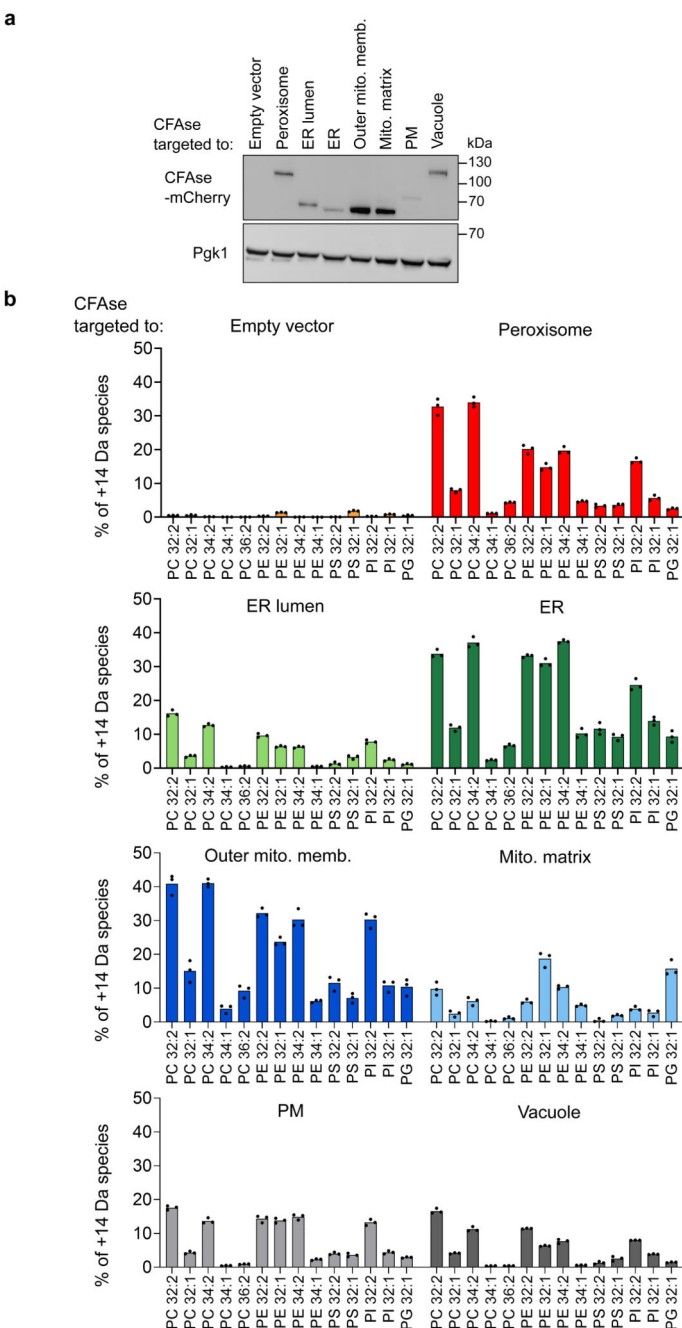

**Figure 2. CFAse mass-tags various phospholipid species in organelles.**

**a,** Western blot depicting levels of mCherry-tagged CFAse when targeted to different organelles; Similar results were obtained in three independent experiments **b,** Bar plot showing the percentage of each indicated phospholipid species that is mass-tagged upon constitutive expression of organelle-targeted CFAse in yeast. Empty vector refers to a plasmid carrying no CFAse coding sequence. Percentage values represent the mean derived from experiments done on three independent clones. Source numerical data and unprocessed blots are available in source data.

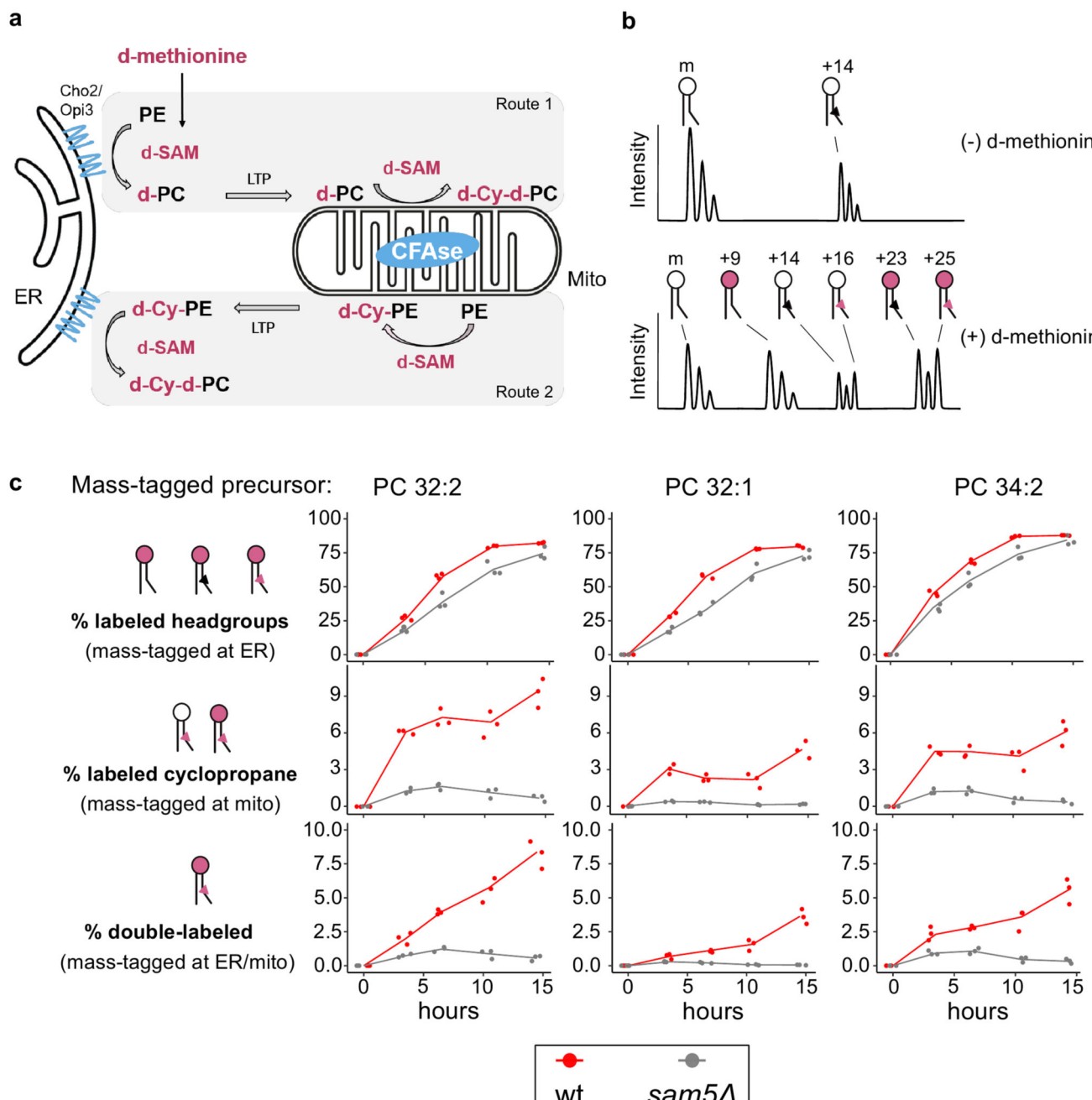

**Figure 3. Monitoring ER-mitochondria lipid exchange using METALIC.**
**a**, CFAse is targeted to the mitochondrial matrix, while the endogenous methyltransferases, Cho2 and Opi3, localize to the ER. These enzymes at the two organelles serve to introduce distinct mass tags. Cells are pulse-labeled with deuterated methionine (d-methionine), resulting in deuterated SAM (d-SAM). In **Route 1**, the first mass-tagging occurs in ER, resulting in d-PC (+9 Da). When d-PC is transported by a lipid transport protein (LTP) to the mitochondrial matrix, a second mass tag (+16 Da) in the form of a deuterated cyclopropane group (d-Cy) is added by CFAse using d-SAM, resulting in d-Cy-d-PC (+25 Da). The

doubly mass-tagged species can also result from **Route 2**, where PE in the mitochondria matrix can get a deuterated cyclopropane mass tag (+16 Da), which subsequently can be double mass-tagged (+9 Da) in the ER by the methyltransferases. **b**, Theoretical mass spectra illustrating the different modifications of a precursor PC species. At steady state, in addition to the precursor, only the cyclopropylated +14 Da species (black triangle) is detected due to the constitutive expression of CFAse. Upon treatment with d-methionine, labeling of the headgroup at the ER (pink headgroups) results in the +9 Da shift. Headgroup labeling of the +14 Da species results in the detection of +23 Da species. Detection of +16 Da species indicates the labeling of the tail in PCs at mitochondria (red triangle). Double mass-labeling of both the headgroup (at ER) and the tail (at mitochondria) result in a +25 Da mass tag. **c**, Line plot depicting the percentage of incorporation in the head group (sum of +9, +23 and +25 species), fatty acid tail (sum of the +16 and +25 species) and both (+25 species) after d-methionine pulse labeling of cells of the indicated genotype at the indicated time points. Three independent clones for each genotype were used. Source numerical data are available in source data.

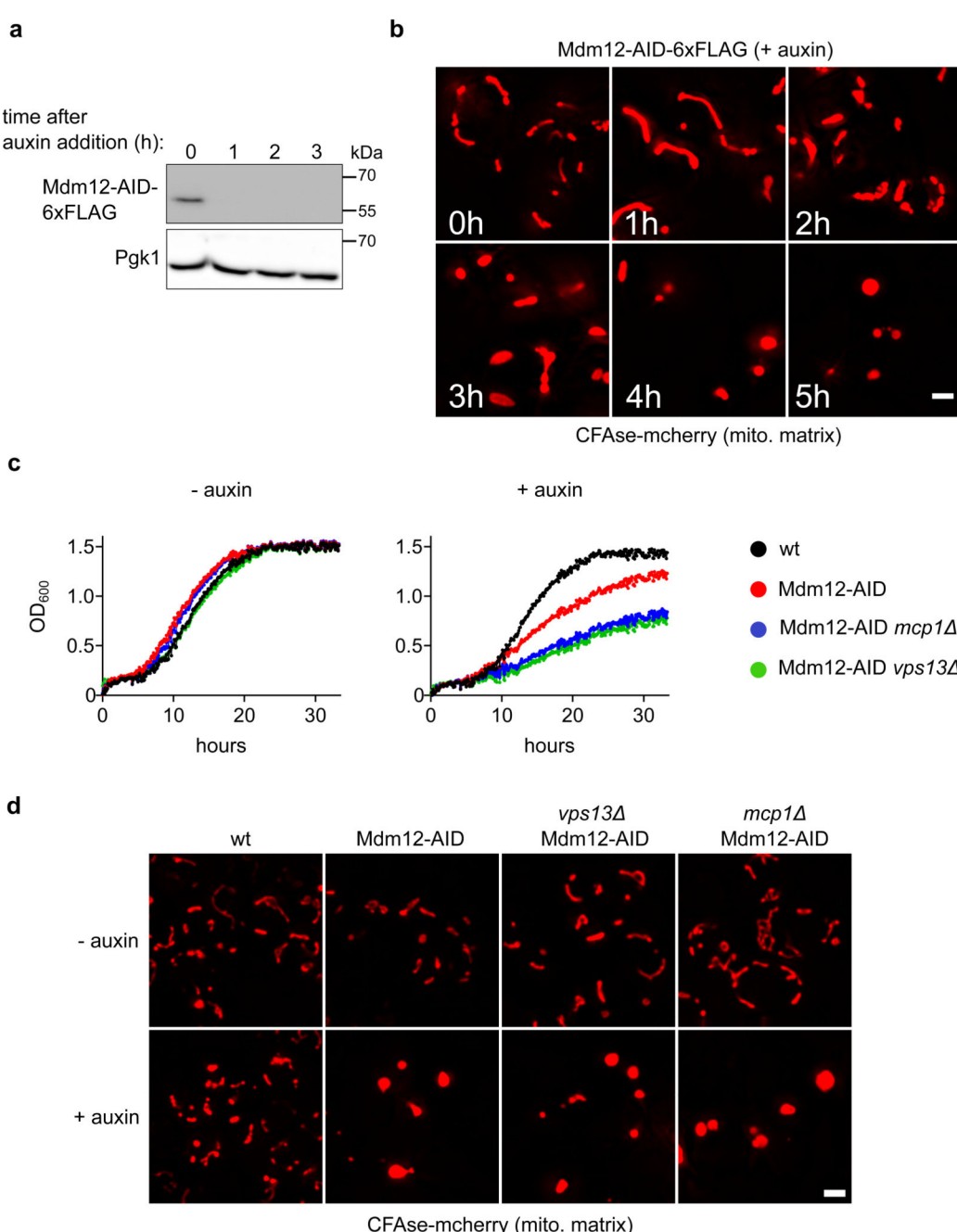

**Figure 4.**

**a**, An auxin-inducible degron (AID) system to inactivate ERMES. Cells co-expressing *At*TIR-9xMyc and Mdm12-AID-6xFLAG were treated with 0.5mM auxin, grown at 30°C and collected at defined time points (hours). Total protein extracts were analyzed by SDS-PAGE and Western blotting. Mdm12 was detected using a α-FLAG antibody. Phosphoglycerate kinase, which serves as a loading control, was detected using a α-PGK antibody; Similar results were obtained in two independent experiments**b**, Cells bearing the Mdm12 degron system and expressing the mitochondria matrix-targeted CFAse were treated

with 0.5mM auxin and imaged at the mentioned time points (hours). Images correspond to a maximum intensity projection of six Z-slices. Similar results were obtained in three independent experiments Scale bar, 2 μm.**c**, Growth of cells with the indicated genotypes was monitored using $OD_{600}$ measurements in the absence or presence of 0.5mM auxin. **d**, Localization of mCherry-tagged CFAse in the indicated strains, either in the absence of auxin or upon treatment with 0.5mM auxin for 7 hours. Similar results were obtained in three independent experiments Scale bar, 2 μm. Source numerical data and unprocessed blots are available in source data.

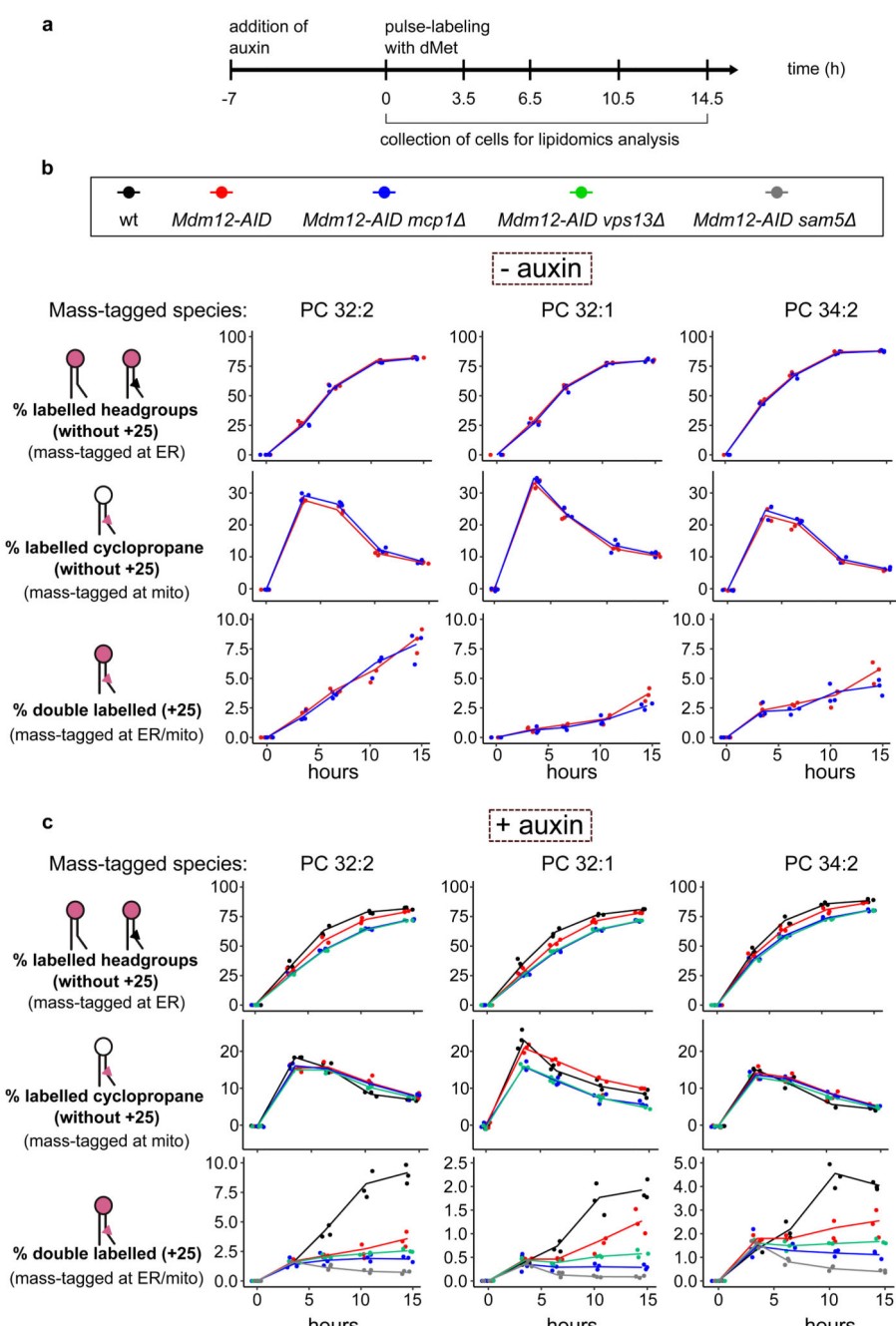

**Figure 5. Kinetics of ER-mitochondria phospholipid exchange.**
**a**, Depiction of the time points at which cells were collected for lipidomics analysis upon pulse labeling with dMet, after treatment with 0.5mM auxin for 7 hours. Line plots showing the fraction of the +9 Da, +16 Da and +25 Da species over time in the indicated genotypes either without (**b**) or with (**c**) auxin treatment. Three independent clones of each genotype were used. Source numerical data are available in source data.

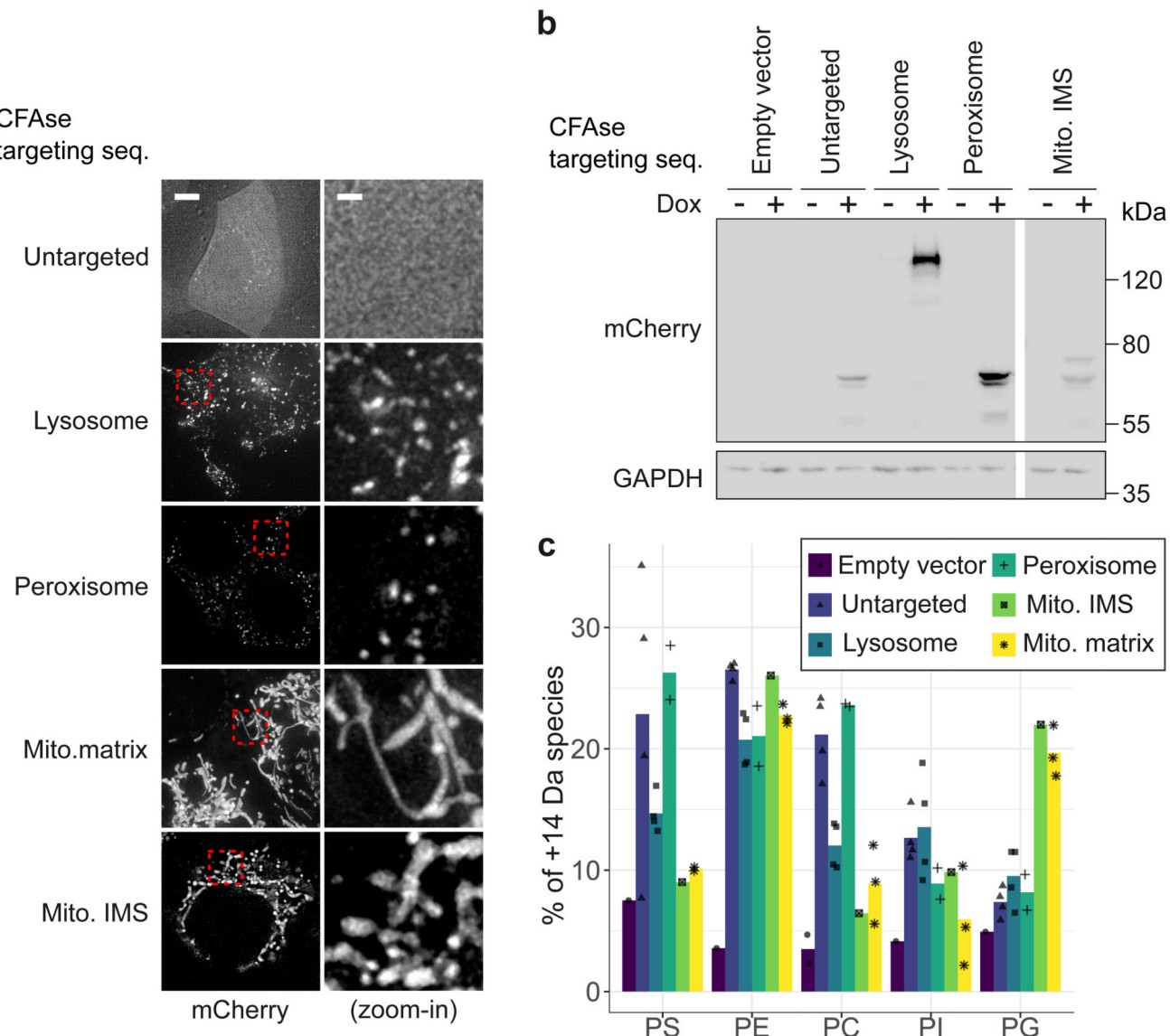

**Figure 6. CFAse mass-tags phospholipids in mammalian cells.**
**a**, Localization of mCherry-tagged CFAse to different organelles by fusion to targeting sequences (Methods). Scale bars, left 5 μm, right 1 μm; Similar results were obtained in at least two independent experiments. **b,** Western blot depicting levels of mCherry-tagged CFAse when targeted to different organelles. This experiment was performed once but similar results were obtained and showed in Extended Data Fig. 5a. **c,** Bar plot showing the percentage of mass-tagged lipid averaging for each detectable species of the indicated phospholipid class upon constitutive expression of organelle-targeted CFAse in HeLa cells. Empty vector refers to a plasmid carrying no CFAse coding sequence. Source numerical data and unprocessed blots are available in source data.

