## [Peer Review File · Nature cell biology]

Peer Review Information

Journal: Nature Cell Biology

Manuscript Title: METALIC reveals interorganelle lipid flux in live cells by enzymatic mass tagging

Corresponding author name(s): Benoît Kornmann

Reviewer Comments & Decisions:

Decision Letter, initial version:

Dear Professor Kornmann,

Thank you for submitting your manuscript, "Interorganelle lipid flux revealed by enzymatic mass tagging in vivo", to Nature Cell Biology. It has now been seen by 3 referees, who are experts in membrane lipids; yeast; organelle contact sites (referee 1); VPS13 (referee 2); lipids; contact sites; has experience with lipidomics (referee 3). As you will see from their comments (attached below), they find this work of potential interest, but have raised substantial concerns, which in our view would need to be addressed with considerable revisions before we can consider publication in Nature Cell Biology.

As you may know, Nature Cell Biology editors discuss the referee reports in detail within the editorial team, including the chief editor, to identify key referee points that should be addressed with priority to strengthen the core conclusions, as opposed to requests that are beyond the scope of the current study. In this case, the reviewers showed enthusiasm and support for the development of an assay allowing the monitoring of inter-organelle lipid fluxes. Rev#2's points could largely be addressed in the text, whereas Rev#1 and #3 request valuable additional analyses and controls to better characterize the method and its potential use across systems. To guide the scope of the revisions, I have listed the points that require focused revision efforts below. We are committed to providing a fair and constructive peer-review process, so please feel free to contact me if you would like to discuss any of the referee comments further. Our typical revision period is six months; however, please let me know if you anticipate any delays, issues addressing the reviews, and/or pandemic-related challenges. We are happy to discuss the revision process and timeline.

In our view, for reconsideration at NCB, it would be essential to:

A. Address Rev#3's comments about the contributions of de novo phospholipid synthesis, and whether perturbing biosynthesis may be informative to interpret transport direction (Rev#3, points #2, #3, #4)

B. Address Rev#1 and Rev#3's questions about the stability of CFAse fatty acids, targeting of CFAse, and substrate availability, which could influence the lipid tagging process (Rev#1 points #1, #4; Rev#3 point #1). Please also consider Rev#1's question in point #5 about the potential action of CFAse in trans at a closely apposed membrane.

C. Address the reviewers' comments about the time scale of the appearance of doubly tagged lipids (Revs#2-3) and the rate of equilibrium in cells compromised for ERMES/Vps13 (Rev#1)

D. Follow up on Rev#1's suggestion to show METALIC could be used in mammalian cells with CFAse targeting (Rev#1 point #2).

E. All other referee concerns pertaining to providing controls, methodological details, clarifications and textual changes should also be addressed. Methodological clarity is especially essential for a Technical Report.

F. Finally please pay close attention to our guidelines on statistical and methodological reporting (listed below) as failure to do so may delay the reconsideration of the revised manuscript. In particular please provide:

We would be happy to consider a revised manuscript that would satisfactorily address these points, unless a similar paper is published elsewhere, or is accepted for publication in Nature Cell Biology in the meantime.

- ensure that it conforms to our format instructions and publication policies (see below and <https://www.nature.com/nature/for-authors>).

- provide a point-by-point rebuttal to the full referee reports verbatim, as provided at the end of this letter.

- provide the completed Reporting Summary (found here <https://www.nature.com/documents/nr-reporting-summary.pdf>). This is essential for reconsideration of the manuscript will be available to editors and referees in the event of peer review. For more information see <http://www.nature.com/authors/policies/availability.html> or contact me.

2When submitting the revised version of your manuscript, please pay close attention to our [href="https://www.nature.com/nature-research/editorial-policies/image-integrity">Digital Image Integrity Guidelines](https://www.nature.com/nature-research/editorial-policies/image-integrity). and to the following points below:

This journal strongly supports public availability of data. Please place the data used in your paper into a public data repository, or alternatively, present the data as Supplementary Information. If data can only be shared on request, please explain why in your Data Availability Statement, and also in the correspondence with your editor. Please note that for some data types, deposition in a public repository is mandatory - more information on our data deposition policies and available repositories appears below.

[REDACTED]

We hope that you will find our referees' comments and editorial guidance helpful. Please do not hesitate to contact me if there is anything you would like to discuss. Thank you again for considering NCB for your work.

Best wishes,

Melina

Melina Casadio, PhD
Senior Editor, Nature Cell Biology
ORCID ID: <https://orcid.org/0000-0003-2389-2243>

Reviewers' Comments:

Reviewer #1:

Remarks to the Author:

This study describes a new technique, called METALIC, for assessing lipid exchange between organelles in cells. The technique relies on localizing a lipid modifying enzyme to various compartments and using the modification as an indirect indication of lipid flux through a compartment. While this type of approach has been used several times before, technical difficulties have limited its general usefulness. METALIC overcomes some of these difficulties. It uses an enzyme, cyclopropane-fatty-acyl phospholipid synthase (CFase), that modifies many classes of phospholipids at once, modified lipids do not seem to be toxic to cells, and the CFase can be easily localized to various cellular compartments. METALIC uses mass spec to assess lipid modification, which allows simultaneous determination of levels of modified and unmodified lipids. There are numerous caveats to the approach. Most are discussed here, but a bit of additional work is necessary to address some of them. Given how challenging it is to assess intracellular lipid flux, METALIC may be a useful new technique. I think a few issues need to be addressed before publication.

1. The stability of CFA fatty acids in *S. cerevisiae* should be addressed. Are CFA fatty acids degraded? What is the half-life of a CFA fatty acid and how does it compare to other fatty acids?
2. To make sure METALIC will be useful for mammalian systems, CFase should be produced in a few types of mammalian cells. It is possible that the polyunsaturated fatty acyl groups found in some phospholipids will form multiple cyclopropyl groups, which could be toxic.
3. The discussion of the ER-mitochondria lipid exchange is confusing. The labeling data should be quantified. Fig. 3C suggests it takes the headgroup labeling of PC about 12-15 hours to come to equilibrium. This should be quantified. It does not seem that the cyclopropane labeling or double labeling of PC has come to equilibrium by 15 hours. This difference should be discussed. The rate of equilibration is even more important to determine for the experiments with Mdm12-AID (Fig. 5 B,C). If ERMES and Vps13 play a significant role in lipid exchange, then the rate of equilibration should slow significantly in cells depleted of these proteins, but that does not seem to be the case. Alternatively, the rate of double labeled PC formation could be determined. There does not seem to be any difference between the strains after 3.5 hours of labeling. This does not seem to support the claim that lipid exchange between the ER and mitochondria slows in cells lacking ERMES and Vps13. Instead, the differences between the strains could indicate the percent of PC that is double labeled at equilibrium is determined by factors other than lipid trafficking.
4. A significant limitation of the METALIC method is that CFase requires SAM, which may not be present in some cellular compartments. This should be discussed. Is there evidence that SAM is in the ER lumen? If not, could the CFA-containing lipids formed in cells with CFase in the ER lumen be formed by a small fraction of the CFase not imported into the ER?
5. When CFase is tethered to the cytoplasm facing side of an organelle is it possible that it can act on

4lipids in a closely apposed membrane (i.e, in trans at contact sites)?

Minor issues:

1. It would be helpful to display the growth curves in Fig. 1D on a semi-log graph and to calculate growth rates. The growth rate of cells expressing untagged CFAse should be added to the figure.

2. A previous study showed it was possible to produce CFA lipids in *S. cerevisiae* and proposed using Raman spectroscopy to study lipid trafficking (PMID: 30207333). This study should be cited and briefly discussed.

Reviewer #2:

Remarks to the Author:

This manuscript from John Peter et al. describes the development of a mass-tagging technique that allows the monitoring of lipid transit between organelles in vivo. The field of intracellular lipid transport has grown extensively with the recent discovery of several different new types of lipid transport proteins. While lipid transport activity can be assayed in vitro, the lack of a versatile in vivo assay is a significant hindrance to the field and so the development of such a technique is an important advance.

The authors' system uses lipid-modifying enzymes targeted to two different organelles. These enzymes alter the mass of the lipids (ie mass-tags them). Transit between the organelles can then be monitored in bulk by using mass spectrometry to follow incorporation of the mass tags. In the proof of principle experiments shown, the bacterial CFAse enzyme was targeted to mitochondria and the second 'tagging enzymes' used were the native, ER localized Cho2 and Opi3 methyltransferases. Both types of tagging enzyme use s-adenosylmethionine as a donor, allowing for pulse-labeling of the lipids with deuterated methionine and the ability to monitor rates of transfer, an important feature for the assay system.

The authors use this system to demonstrate for the first time that the ERMES complex and the Vps13/Mcp1 proteins play redundant roles in the transfer of lipids between the mitochondrial and the ER, a result that had been inferred from genetic studies.

The data presented are of high quality and the paper is clearly written and accessible.

There are several limitations to this current version of the system that are noted by the authors: the direction of transfer can't be established, whether the transfer is direct or involves transit through intermediate compartments is not clear, and only the transit of lipids between different organelles and the ER can be assayed. Nonetheless, the establishment of a system for in vivo assay of lipid transport is important and, undoubtedly, further refinements will address some of these deficiencies.

Major comments

The time scale of the appearance of doubly tagged lipids (Figure 3C) is extremely slow – on the scale

5of multiple cell cycles. Presumably this doesn't reflect the true rate of transfer in vivo. This requires some comment in the text. Is this a technical issue?

The senior authors' lab has previously reported that Vps13/Mcp1 influence mitochondrial lipids through mitochondrial-vacuole junctions. The authors note that the current assay can't distinguish between direct transfer between the ER and the mitochondria and transfer that involves movement through intermediary organelles. Nonetheless, it should be clearly stated that the redundancy shown here for ERMES and Vps13/Mcp1 in shuttling lipids between the ER and Mitochondria most likely reflects different routes for the lipids in the two cases.

A clearer explanation of why the activity of the methyltransferases produces a shift of 9 mass units would be helpful. It took a while to work that out.

Reviewer #3:

Remarks to the Author:

This very interesting technical report develops a mass-tag labeling system in yeast to monitor the interorganelle transport of lipids. The study demonstrates it can target the CFAse enzyme to different organelles, and that it can monitor the lipid modifications created from enzymes CFAse and methyltransferase on lipids with mass-spec lipidomics. This in essence creates a platform to monitor interorganelle lipid trafficking in vivo by monitoring the incorporation of two lipid modifications at different organelles. As a proof of principle, they then examine how loss of the ERMES complex and Vps13/Mcp1 machinery impact ER-mitochondria double lipid labeling.

The work develops a new assay and begins to address a long-standing question in organelle lipid transport biology, namely how do ERMES and Vps13/Mcp1 contribute to non-vesicular lipid transport. The tools developed here are creative and innovative, and it is clear there is potential to utilize them to monitor lipid transport between several organelles. However, there are several controls that need to be added to provide rigour to the study.

Major issues:

1. The first issue deals with experiments where CFAse is targeted to different organelles. The enzyme can clearly be targeted as shown by microscopy, and targeting to different organelles shows similar patterns of relative abundances of labeled lipids for most organelles. However there are differences in the % of 14-Da labeled lipids in different compartments. For example ER lumen labeling is significantly less than outer mito labeling. It is unclear why this differential labeling occurs. Is the CFAse enzyme abundance different in the different organelles? Some Western blotting to show how organelle targeting relates to CFAse abundance would help clarify some of this.

2. A second issue is whether changes in the rate of de novo phospholipid synthesis impact some experiments. Figure 5 beautifully shows that over time, there is increased detection of headgroup labeled PC at the ER, consistent with its labeling at its organelle of synthesis. Although the rates are comparable, there is slightly less headgroup labeled PC in the MDM12-AID mcp1/vps13 strains (Figure

65C). What are the rates of de novo PC synthesis in these backgrounds? This seems very pertinent as ER-mito contacts have been previously implicated as key sites for de novo phospholipid synthesis in the past. Decreased lipid synthesis in the ERMES/Vps13-minus background may over-estimate the efficiency of ER-localized labeling, since there is less unlabeled lipid being constantly made to dilute our labeled lipid.

3. Related to point 2, would the rate of ER headgroup labeling increase if de novo lipogenesis were completely halted (which is constantly supplying new unlabeled lipids to the system), such as with addition of cerulenin?

4. Related to the direction of the lipid transport (route 1 vs 2): would perturbing ER lipid synthesis tell you anything about the directionality of the interorganelle lipid flow? Halting ER lipid synthesis may impact ER-to-mito more than mito-to-ER?

5. A third point concerns the time scales of the experiments. These experiments are conducted on the scale of several hours, but the abundance of double-labeled lipids is very low by the end. In particular, the % of incorporation of the double labeled (+25 species) in Figure 3 experiments is only about 10% or less. Based on the expected lipid trafficking of lipids between ER and mitochondria, this seems like a low level of trafficked lipid compared to what lipids would, in theory, be able to traffick between these compartments at this time scale. More comment on this in the Discussion would be very helpful.

7AUTHOR NAMES – should be given in full.

Methods should be written concisely, but should contain all elements necessary to allow interpretation and replication of the results. As a guideline, Methods sections typically do not exceed 3,000 words.

The Methods should be divided into subsections listing reagents and techniques. When citing previous methods, accurate references should be provided and any alterations should be noted. Information must be provided about: antibody dilutions, company names, catalogue numbers and clone numbers for monoclonal antibodies; sequences of RNAi and cDNA probes/primers or company names and catalogue numbers if reagents are commercial; cell line names, sources and information on cell line identity and authentication. Animal studies and experiments involving human subjects must be reported in detail, identifying the committees approving the protocols. For studies involving human subjects/samples, a statement must be included confirming that informed consent was obtained. Statistical analyses and information on the reproducibility of experimental results should be provided in a section titled "Statistics and Reproducibility".

All Nature Cell Biology manuscripts submitted on or after March 21 2016 must include a Data availability statement as a separate section after Methods but before references, under the heading "Data Availability". For Springer Nature policies on data availability see <http://www.nature.com/authors/policies/availability.html>; for more information on this particular policy see <http://www.nature.com/authors/policies/data/data-availability-statements-data-citations.pdf>. The Data availability statement should include:

- Accession codes for primary datasets (generated during the study under consideration and designated as "primary accessions") and secondary datasets (published datasets reanalysed during the study under consideration, designated as "referenced accessions"). For primary accessions data should be made public to coincide with publication of the manuscript. A list of data types for which submission to community-endorsed public repositories is mandated (including sequence, structure, microarray, deep sequencing data) can be found here <http://www.nature.com/authors/policies/availability.html#data>.
- Unique identifiers (accession codes, DOIs or other unique persistent identifier) and hyperlinks for datasets deposited in an approved repository, but for which data deposition is not mandated (see here for details <http://www.nature.com/sdata/data-policies/repositories>).
- At a minimum, please include a statement confirming that all relevant data are available from the authors, and/or are included with the manuscript (e.g. as source data or supplementary information), listing which data are included (e.g. by figure panels and data types) and mentioning any restrictions on availability.
- If a dataset has a Digital Object Identifier (DOI) as its unique identifier, we strongly encourage including this in the Reference list and citing the dataset in the Methods.

We recommend that you upload the step-by-step protocols used in this manuscript to the Protocol Exchange. More details can found at www.nature.com/protocolexchange/about.

9FIGURES – Colour figure publication costs \$600 for the first, and \$300 for each subsequent colour figure. All panels of a multi-panel figure must be logically connected and arranged as they would appear in the final version. Unnecessary figures and figure panels should be avoided (e.g. data presented in small tables could be stated briefly in the text instead).

All imaging data should be accompanied by scale bars, which should be defined in the legend. Cropped images of gels/blots are acceptable, but need to be accompanied by size markers, and to retain visible background signal within the linear range (i.e. should not be saturated). The boundaries of panels with low background have to be demarked with black lines. Splicing of panels should only be considered if unavoidable, and must be clearly marked on the figure, and noted in the legend with a statement on whether the samples were obtained and processed simultaneously. Quantitative comparisons between samples on different gels/blots are discouraged; if this is unavoidable, it should only be performed for samples derived from the same experiment with gels/blots were processed in parallel, which needs to be stated in the legend.

- Some programs can generate Postscript by 'printing to file' (found in the Print dialogue). If using an application not listed above, save the file in PostScript format or email our Art Editor, Allen Beattie for

10advice (a.beattie@nature.com).

The total number of Supplementary Figures (not including the “unprocessed scans” Supplementary

11Figure) should not exceed the number of main display items (figures and/or tables (see our Guide to Authors and March 2012 editorial <http://www.nature.com/ncb/authors/submit/index.html#suppinfo>; <http://www.nature.com/ncb/journal/v14/n3/index.html#ed>). No restrictions apply to Supplementary Tables or Videos, but we advise authors to be selective in including supplemental data.

GUIDELINES FOR EXPERIMENTAL AND STATISTICAL REPORTING

REPORTING REQUIREMENTS – We are trying to improve the quality of methods and statistics reporting in our papers. To that end, we are now asking authors to complete a reporting summary that collects information on experimental design and reagents. The Reporting Summary can be found here <https://www.nature.com/documents/nr-reporting-summary.pdf>) If you would like to reference the guidance text as you complete the template, please access these flattened versions at <http://www.nature.com/authors/policies/availability.html>.

Author Rebuttal to Initial comments

We take the occasion to thank the reviewers for their constructive criticism and useful suggestions, which allow us to come back to you today with a much stronger story.

Please find our specific responses below.

Reviewer #1:

1. The stability of CFA fatty acids in *S. cerevisiae* should address. Are CFA fatty acids degraded? What is the half-life of a CFA fatty acid and how does it compare to other fatty acids?

We have addressed this question by feeding yeast a mixture of cyclopropyl fatty acid and deuterated oleic acid. Both fatty acids can be distinguished from endogenously produced fatty-acids by mass spectrometry. We then starved cells for carbon, to a)-prevent them from diluting the fed fatty-acids with newly-synthesized fatty-acids, b)-force them to metabolize their fatty acids. We then followed the rate of disappearance of free cyclopropyl and deuterated Fatty-acids, and found no difference in their turnover. From this we conclude that CFA fatty acids are degraded with the same kinetics as other fatty acids.

2. To make sure METALIC will be useful for mammalian systems, CFase should be produced in a few types of mammalian cells. It is possible that the polyunsaturated fatty acyl groups found in some phospholipids will form multiple cyclopropyl groups, which could be toxic.

We show that CFase can be transfected in HeLa cells, targeted to various organelles, and produce detectable amounts of CFA lipids without noticeable effect on cell survival. These data are very promising for the use of METALIC in mammalian cells (New fig 6). One caveat that we note in the manuscript, however, is that mammalian cells generate a small amount of ether-bound plasmalogens, the m/z of which is identical to that of cyclopropylated lipids. Therefore, the amounts measured in mock transfections appear non-zero. This is however not problematic as this background signal is small, and can be entirely eliminated using tandem mass-spectrometry. Indeed, if the precursors have identical m/z , it is not the case for the fragmentation products. We think that the data we provide here are

satisfactory and sufficient to provide the proof-of-principle demanded. We think that performing additional analyses would be beyond the scope of our manuscript.

We also cite studies showing that *C. elegans* incorporate and benefit from dietary cyclopropane fatty acids from the *E. coli* they feed on (Kaul TK et al. PLoS One 2014, Castro C et al. BMC genomics 2012). Thus, provided that *C. elegans* are fed CFAse mutant *E. coli*, METALIC should be amenable to this invertebrate system as well without adverse effects.

3. The discussion of the ER-mitochondria lipid exchange is confusing. The labeling data should be quantified. Fig. 3C suggests it takes the headgroup labeling of PC about 12-15 hours to come to equilibrium. This should be quantified. It does not seem that the cyclopropane labeling or double labeling of PC has come to equilibrium by 15 hours. This difference should be discussed. The rate of equilibration is even more important to determine for the experiments with Mdm12-AID (Fig. 5 B,C). If ERMES and Vps13 play a significant role in lipid exchange, then the rate of equilibration should slow significantly in cells depleted of these of these proteins, but that does not seem to be the case. Alternatively, the rate of double labeled PC formation could be determined. There does not seem to be any difference between the strains after 3.5 hours of labeling. This does not seem to support the claim that lipid exchange between the ER and mitochondria slows in cells lacking ERMES and Vps13. Instead, the differences between the strains could indicate the percent of PC that is double labeled at equilibrium is determined by factors other than lipid trafficking.

We apologize for the confusion. The labelling does not, in fact, come to equilibrium, but new label incorporation continuously dilutes non-labelled lipid species asymptotically. For headgroups, labelling approaches 100% as cells grow to dilute non-labeled lipids. For CFA lipids, labelling approaches ~10, ~3 and ~6% for PC32:2, PC32:1 and PC34:2 (Fig 3C), respectively, as this is the final proportion of labelled lipid by the matrix-directed CFAse (Fig 2).

It is an important question indeed as to whether the rates we observe are within the range that could be expected. We have therefore compared our observations to theoretical predictions from a model in which newly synthesized labeled lipids dilute existing non-labeled lipids. We find that our observations are in fact quite close to expectations. The new Extended data Fig 3B shows the expected label incorporation (to be compared to figure 3C top panels). New Extended data Fig 3C-E shows the ratio of observed vs expected labelling, where measurable (i.e. not at t_0 nor in Sam5 mutants for CFA labelling). We find that the ratio is always close to 1, indicating good agreement between theory and observation. We also now explicitly address the discrepant data that we observe at $t_{3.5}$, even in our negative control. Addition of excess methionine at t_0 is likely to cause important physiological changes, e.g. a burst in SAM accumulation before methionine transporters are downregulated by the arrestin pathway, allowing

14labelling kinetics to reach a steady-state. We therefore think that later timepoints are less prone to be influenced by physiological changes induced by methionine addition to yeast cells. We clarified this point in the text.

4. A significant limitation of the METALIC method is that CFAse requires SAM, which may not be present in some cellular compartments. This should be discussed. Is there evidence that SAM is in the ER lumen? If not, could the CFA-containing lipids formed in cells with CFAse in the ER lumen be formed by a small fraction of the CFAse not imported into the ER?

This is an important limitation that we now fully acknowledge (p. 16). While we do not know how much SAM there is in the ER lumen, two permeases ferry SAM across membranes of the endomembrane system, one being the ER-localized Sam3, the other being uncharacterized (Rouillon A et al. JBC 1999). Deletion of SAM3 had no effect on the labelling efficiency of ER-localized CFAse (not shown), preventing us to use it as a control, like we use SAM5 for mitochondrial matrix.

We can nevertheless exclude the possibility that the cyclopropylation by ER-lumen-directed CFAse is due to a small fraction of non-imported protein. Indeed, if it were the case, then the lipid specificity of ER-lumen-directed CFAse should be identical to that of untargeted CFAse. We have therefore measured the specificity of cytosolic CFAse. New Extended data Fig 2 shows marked differences in specificity between untargeted and ER-lumen directed CFAse when the overall efficiency is normalized (i.e. plotted on two axes), indicating that the detectable activity of the ER-lumen-directed CFAse is not due to a small fraction of non-translocated protein

This is all now clarified and discussed in the text (p. 6)..

5. When CFAse is tethered to the cytoplasm facing side of an organelle is it possible that it can act on lipids in a closely apposed membrane (i.e, in trans at contact sites)?

This is an important possibility that cannot be excluded and that we now duly acknowledge in the discussion section (p. 16).

Minor issues:

1. It would be helpful to display the growth curves in Fig. 1D on a semi-log graph and to calculate growth rates. The growth rate of cells expressing untagged CFAse should be added to the figure.

Done.

2. A previous study showed it was possible to produce CFA lipids in *S. cerevisiae* and proposed using Raman spectroscopy to study lipid trafficking (PMID: 30207333). This study should be cited and briefly discussed.

Thanks for pointing us to this study that we had overlooked and that we now duly discuss (p. 16).

Reviewer #2:

The time scale of the appearance of doubly tagged lipids (Figure 3C) is extremely slow – on the scale of multiple cell cycles. Presumably this doesn't reflect the true rate of transfer in vivo. This requires some comment in the text. Is this a technical issue?

Please also see response to Reviewer #1, point 3. We have compared our observations to theoretical expectations from a model in which newly-synthesized lipids dilute existing non-labelled lipids, with no lipid turnover (a reasonable assumption in conditions where cells do not have any nutrient limitation). We find that our observations are in fact quite close to expectations. The new Extended data Fig 3B shows the expected label incorporation (to be compared to figure 3C top panels). New Extended data Fig 3C-E shows the ratio of observed vs expected labelling, where measurable (i.e. not at t_0 nor in Sam5 mutants for CFA labelling). We find that the ratio is always close to 1, indicating good agreement between theory and observation. This is true for the appearance of labeled headgroups, CFAs, and importantly of double-labelled species, where the theoretical levels are calculated as

(Fraction of headgroup-labelled lipids) x (Fraction of CFA-labelled lipids).

We now explain in the text that our data are compatible with a no-turnover model.

The senior authors' lab has previously reported that Vps13/Mcp1 influence mitochondrial lipids through mitochondrial-vacuole junctions. The authors note that the current assay can't distinguish between direct transfer between the ER and the mitochondria and transfer that involves movement through intermediary organelles. Nonetheless, it should be clearly stated that the redundancy shown here for ERMES and Vps13/Mcp1 in shuttling lipids between the ER and Mitochondria most likely reflects different routes for the lipids in the two cases.

16This is indeed a great point of discussion, which we hadn't expanded upon, as we were likely too focused on the technical aspect and not enough on their biological implications. This is now rectified in two places on p. 16.

A clearer explanation of why the activity of the methyltransferases produces a shift of 9 mass units would be helpful. It took a while to work that out.

Done (p.8).

Reviewer #3:

Major issues:

1. The first issue deals with experiments where CFAse is targeted to different organelles. The enzyme can clearly be targeted as shown by microscopy, and targeting to different organelles shows similar patterns of relative abundances of labeled lipids for most organelles. However there are differences in the % of 14-Da labeled lipids in different compartments. For example ER lumen labeling is significantly less than outer mito labeling. It is unclear why this differential labeling occurs. Is the CFAse enzyme abundance different in the different organelles? Some Western blotting to show how organelle targeting relates to CFAse abundance would help clarify some of this.

We now provide western-blot quantification of the CFAse levels (new Fig 2A). Variation of CFAse levels do not necessarily explain differences in cyclopropylation which can also be due to variations in SAM concentration in the target compartment, substrate abundance or any other particularities of that compartment. We clarified this in the text (end of p. 6).

2. A second issue is whether changes in the rate of de novo phospholipid synthesis impact some experiments. Figure 5 beautifully shows that over time, there is increased detection of headgroup labeled PC at the ER, consistent with its labeling at its organelle of synthesis. Although the rates are comparable, there is slightly less headgroup labeled PC in the MDM12-AID mcp1/vps13 strains (Figure 5C). What are the rates of de novo PC synthesis in these backgrounds? This seems very pertinent as ER-mito contacts have been previously implicated as key sites for de novo phospholipid synthesis in the past. Decreased lipid synthesis in the ERMES/Vps13-minus background may over-estimate the efficiency of ER-localized labeling, since there is less unlabeled lipid being constantly made to dilute our labeled lipid.

The rate of PS and PE biosynthesis in *ERMES* mutants has been measured in Nguyen TT et al. Traffic 2012 and Lang AB et al. J Cell Biol 2015. *ERMES* mutants and double *ERMES/Vps13* mutants show little to no decrease in PS and PE biosynthesis rates. Because we only measure PC, there is no “unlabeled lipid being constantly made to dilute our labeled lipid”. All newly-synthesized lipids are labelled. This is consistent with the expectations of the no-turnover model (see responses to Reviewer #1 point 3 and Reviewer #2, point 1).

3. Related to point 2, would the rate of ER headgroup labeling increase if *de novo* lipogenesis were completely halted (which is constantly supplying new unlabeled lipids to the system), such as with addition of cerulenin?

Related to Reviewer #1 point 3 and Reviewer #2 point 1, our data indicate that the rate of headgroup labelling is in fact in line with the predictions of a model in which label is incorporated into newly synthesized lipids, diluting the pool of pre-existing unlabelled lipids, with little or no lipid turnover. Cerulenin treatment would likely inhibit cell growth and thus the incorporation of label into newly synthesized lipids. We focus our analysis on PC. PC is the labelled lipid; the unlabelled precursor would be PE. It is questionable whether, if lipid synthesis is inhibited, all PE would be turned to PC. Feedback mechanism might instead regulate the activity of enzymes to prevent that from happening.

4. Related to the direction of the lipid transport (route 1 vs 2): would perturbing ER lipid synthesis tell you anything about the directionality of the interorganelle lipid flow? Halting ER lipid synthesis may impact ER-to-mito more than mito-to-ER?

For the reasons exposed in the point above, it is unclear what the effect of perturbing lipid biosynthesis would be. Preventing ER lipid biosynthesis would prevent cell growth and thus labelling of newly-synthesized lipids. We are therefore not optimistic that this approach could help in determining directionality as of now.

5. A third point concerns the time scales of the experiments. These experiments are conducted on the scale of several hours, but the abundance of double-labeled lipids is very low by the end. In particular, the % of incorporation of the double labeled (+25 species) in Figure 3 experiments is only about 10% or less. Based on the expected lipid trafficking of lipids between ER and mitochondria, this seems like a low level of trafficked lipid compared to what lipids would, in theory, be able to traffick between these compartments at this time scale. More comment on this in the Discussion would be very helpful.

Related to Reviewer #1 point 3 and Reviewer #2 point 1, our data indicate that the rates of labelling are in line with the predictions of a “no turnover” model of lipid synthesis. The ~10%, ~3% and ~6% labelling observed for PC32:2, PC32:1 and PC34:2, respectively, are in fact in line with the steady-state percentage of CFA modification observed in figure 2 for matrix-targeted CFase. The abundance of double-labelled lipid cannot raise above these levels.

Decision Letter, first revision:

4th March 2022

Dear Dr. Kornmann,

Thank you for submitting your revised manuscript "Interorganelle lipid flux revealed by enzymatic mass tagging in vivo" (NCB-K46467A). It has now been seen by the original referees and their comments are below. The reviewers find that the paper has improved in revision, and therefore we'll be happy in principle to publish it in Nature Cell Biology, pending minor revisions to satisfy the referees' final requests and to comply with our editorial and formatting guidelines.

You will see that Rev#1 had remaining issues related to data interpretation and limits of the approach. Their concerns are persistent and consistent with the previous round of review. We agree with the reviewer and feel that their point #1 should be further addressed textually. We discussed their second point with Rev#3 (please see additional comments from Rev#3 below their review comments). We understand that there are differences in interpretation amongst these experts and strongly encourage you to address point #2 to the best of your ability; if you can add additional controls, please feel free to add them, and we look forward to your thoughts on these points.

We will now start performing detailed checks on your paper and will send you a checklist detailing our editorial and formatting requirements in about a week. Please do not upload the final materials and make any revisions to the manuscript text/figures until you receive this additional information from us.

Thank you again for your interest in Nature Cell Biology Please do not hesitate to contact me if you have any questions.

Sincerely,

19Melina

Melina Casadio, PhD
Senior Editor, Nature Cell Biology
ORCID ID: <https://orcid.org/0000-0003-2389-2243>

Reviewer #1 (Remarks to the Author):

This study has been significantly improved, but there are still some concerns.

1. The amount of SAM in extra-cytoplasmic compartments remains an important issue and could limit the usefulness of METALIC as a tool to assess lipid flux to compartments other than mitochondria, which are known to import SAM. All known SAM-requiring methyltransferases in *S. cerevisiae* (PMID: 21858014) have active sites in the cytoplasm, nucleus, or mitochondria. This suggests cells do not require SAM outside mitochondria and the cytoplasm and, as far as I know, there is little evidence SAM is there. SAM could well be present in extra-cytoplasmic compartments at levels far below the Km of CFAse. If this is correct, SAM levels, rather than lipid flux, could be the primary determinant of CFA formation when CFAse is outside mitochondria or the cytoplasm. This is a significant limitation on the general usefulness of METALIC. As it stands now, METALIC will certainly be useful for assessing lipid flux between mitochondria and the rest of the cell. For other compartments, using METALIC will require measurements of SAM levels in organelles, which is not trivial. One way around this is to localize CFAse on the cytoplasmic surface of organelles, but this will require determination of the stability of CFAse-containing fusions to rule out that they do not release free CFAse into the cytoplasm. These are not insurmountable problems, but they should be discussed more.

2. I remain perplexed by the authors' interpretation of the findings in Fig. 5. The timing of formation of the double labeled PC species does not seem to support the claim that Mdm12 and Vps13 make significant contributions to lipid flux between mitochondria and the rest of the cell. If they did, then surely auxin treatment would cause a significant reduction in double labeled PC levels 3.5 hours after labeling started (the first time point). The fact that none is found cannot be explained by differences in growth rate, since the cells are still growing at the about the same rate when the 3.5-hour sample was taken (Fig. 4c). To me, this suggests none of the proteins tested makes a significant contribution. What happens at later times could be caused by factors other than changes in lipid flux, such as a decrease in SAM levels. All this is not to say that METALIC is not an interesting new method to study lipid flux, but Fig. 5 is not a strong proof of concept.

Reviewer #2 (Remarks to the Author):

The authors have satisfactorily dealt with all my concerns.

20Reviewer #3 (Remarks to the Author):

The revised manuscript addresses the majority of concerns raised. The new Western blotting in Fig 2 makes it more clear how CFase organelle targeting influences enzyme abundance. The other major concerns have been addressed via comparisons to theoretical modeling of lipid turnover. Finally the addition of new data with CFase targeted to multiple mammalian organelles adds to the impact of the study.

COMMENTS ADDED DURING CROSS-COMMENTING, weighing in on Rev#1's points:

Regarding point 1, I agree that at present it makes sense to address this potential issue with more commentary in the discussion. The issue with SAM does potentially limit the general use of METALIC for certain organelles, and this limitation should be very clearly stated and discussed more.

Regarding point 2...I see the concern here, however, my interpretation of the Fig 5C data was the 3.5hr plot was the yeast were still in their early growth phase of the experiment, and still coming out of their initial lag phase when transferred to the initial culture in the presence of auxin. Generally when inoculating yeast into fresh cultures for experiments like this, one may see the first few hours of the culture as in an "initial lag" phase as they prepare for logarithmic growth. This seems the case in Fig 4C, as their initial growth rate is slower at ~3.5hrs (all are equally slower, as noted by Rev1). At this time, initial membrane synthesis may be somewhat more supported from preexisting lipid stores like those in lipid droplets (and thus may not rely as extensively on inter-organelle transporters like Vps13). As growth rate picks up, this may change.

In any case, I think it may be something the authors should at least address by discussion. I struggle to think of better controls. I am curious what they will say regarding point 2, but as for me, I took that it had more to do with the initial yeast lag phase.

Our ref: NCB-K46467A

21st March 2022

Dear Dr. Kornmann,

Thank you for your patience as we've prepared the guidelines for final submission of your Nature Cell Biology manuscript, "Interorganelle lipid flux revealed by enzymatic mass tagging in vivo" (NCB-K46467A). Please carefully follow the step-by-step instructions provided in the attached file, and add a response in each row of the table to indicate the changes that you have made. Please also check and comment on any additional marked-up edits we have proposed within the text. Ensuring that each point is addressed will help to ensure that your revised manuscript can be swiftly handed over to our production team.

We would like to start working on your revised paper, with all of the requested files and forms, as

21soon as possible (preferably within one week). Please get in contact with us if you anticipate delays.

In recognition of the time and expertise our reviewers provide to Nature Cell Biology's editorial process, we would like to formally acknowledge their contribution to the external peer review of your manuscript entitled "Interorganelle lipid flux revealed by enzymatic mass tagging in vivo". For those reviewers who give their assent, we will be publishing their names alongside the published article.

Nature Cell Biology offers a Transparent Peer Review option for new original research manuscripts submitted after December 1st, 2019. As part of this initiative, we encourage our authors to support increased transparency into the peer review process by agreeing to have the reviewer comments, author rebuttal letters, and editorial decision letters published as a Supplementary item. When you submit your final files please clearly state in your cover letter whether or not you would like to participate in this initiative. Please note that failure to state your preference will result in delays in accepting your manuscript for publication.

Cover suggestions

As you prepare your final files we encourage you to consider whether you have any images or illustrations that may be appropriate for use on the cover of Nature Cell Biology.

Nature Cell Biology has now transitioned to a unified Rights Collection system which will allow our Author Services team to quickly and easily collect the rights and permissions required to publish your work. Approximately 10 days after your paper is formally accepted, you will receive an email in

22providing you with a link to complete the grant of rights. If your paper is eligible for Open Access, our Author Services team will also be in touch regarding any additional information that may be required to arrange payment for your article.

Please note that *Nature Cell Biology* is a Transformative Journal (TJ). Authors may publish their research with us through the traditional subscription access route or make their paper immediately open access through payment of an article-processing charge (APC). Authors will not be required to make a final decision about access to their article until it has been accepted. Find out more about Transformative Journals

Please use the following link for uploading these materials:
[REDACTED]

Best regards,

Nyx Hills
Staff
Nature Cell Biology

On behalf of

Melina Casadio, PhD
Senior Editor, Nature Cell Biology
ORCID ID: <https://orcid.org/0000-0003-2389-2243>

23Reviewer #1:

Remarks to the Author:

This study has been significantly improved, but there are still some concerns.

1. The amount of SAM in extra-cytoplasmic compartments remains an important issue and could limit the usefulness of METALIC as a tool to assess lipid flux to compartments other than mitochondria, which are known to import SAM. All known SAM-requiring methyltransferases in *S. cerevisiae* (PMID: 21858014) have active sites in the cytoplasm, nucleus, or mitochondria. This suggests cells do not require SAM outside mitochondria and the cytoplasm and, as far as I know, there is little evidence SAM is there. SAM could well be present in extra-cytoplasmic compartments at levels far below the K_m of CFAse. If this is correct, SAM levels, rather than lipid flux, could be the primary determinant of CFA formation when CFAse is outside mitochondria or the cytoplasm. This is a significant limitation on the general usefulness of METALIC. As it stands now, METALIC will certainly be useful for assessing lipid flux between mitochondria and the rest of the cell. For other compartments, using METALIC will require measurements of SAM levels in organelles, which is not trivial. One way around this is to localize CFAse on the cytoplasmic surface of organelles, but this will require determination of the stability of CFAse-containing fusions to rule out that they do not release free CFAse into the cytoplasm. These are not insurmountable problems, but they should be discussed more.

2. I remain perplexed by the authors' interpretation of the findings in Fig. 5. The timing of formation of the double labeled PC species does not seem to support the claim that Mdm12 and Vps13 make significant contributions to lipid flux between mitochondria and the rest of the cell. If they did, then surely auxin treatment would cause a significant reduction in double labeled PC levels 3.5 hours after labeling started (the first time point). The fact that none is found cannot be explained by differences in growth rate, since the cells are still growing at the about the same rate when the 3.5-hour sample was taken (Fig. 4c). To me, this suggests none of the proteins tested makes a significant contribution. What happens at later times could be caused by factors other than changes in lipid flux, such as a decrease in SAM levels. All this is not to say that METALIC is not an interesting new method to study lipid flux, but Fig. 5 is not a strong proof of concept.

Reviewer #2:

Remarks to the Author:

The authors have satisfactorily dealt with all my concerns.

Reviewer #3:

Remarks to the Author:

The revised manuscript addresses the majority of concerns raised. The new Western blotting in Fig 2 makes it more clear how CFAse organelle targeting influences enzyme abundance. The other major

24concerns have been addressed via comparisons to theoretical modeling of lipid turnover. Finally the addition of new data with CFase targeted to multiple mammalian organelles adds to the impact of the study.

COMMENTS ADDED DURING CROSS-COMMENTING, weighing in on Rev#1's points:

Regarding point 1, I agree that at present it makes sense to address this potential issue with more commentary in the discussion. The issue with SAM does potentially limit the general use of METALIC for certain organelles, and this limitation should be very clearly stated and discussed more.

Regarding point 2...I see the concern here, however, my interpretation of the Fig 5C data was the 3.5hr plot was the yeast were still in their early growth phase of the experiment, and still coming out of their initial lag phase when transferred to the initial culture in the presence of auxin. Generally when inoculating yeast into fresh cultures for experiments like this, one may see the first few hours of the culture as in an "initial lag" phase as they prepare for logarithmic growth. This seems the case in Fig 4C, as their initial growth rate is slower at ~3.5hrs (all are equally slower, as noted by Rev1). At this time, initial membrane synthesis may be somewhat more supported from preexisting lipid stores like those in lipid droplets (and thus may not rely as extensively on inter-organelle transporters like Vps13). As growth rate picks up, this may change.

In any case, I think it may be something the authors should at least address by discussion. I struggle to think of better controls. I am curious what they will say regarding point 2, but as for me, I took that it had more to do with the initial yeast lag phase.

Author Rebuttal, first revision:

Dear Editor,

We are happy to resubmit our manuscript "*Interorganelle lipid flux revealed by enzymatic mass tagging in vivo*" in which we address the remaining concerns of the reviewers. We would like to thank all the reviewers and appreciate their constructive criticisms and suggestions, which have helped to improve the manuscript substantially.

Reviewer #1 (Remarks to the Author):

This study has been significantly improved, but there are still some concerns.

1. The amount of SAM in extra-cytoplasmic compartments remains an important issue and could limit the usefulness of METALIC as a tool to assess lipid flux to compartments other than mitochondria, which are known to import SAM. All known SAM-requiring methyltransferases in *S. cerevisiae* (PMID: 21858014) have active sites in the cytoplasm, nucleus, or mitochondria. This suggests cells do not

25require SAM outside mitochondria and the cytoplasm and, as far as I know, there is little evidence SAM is there. SAM could well be present in extra-cytoplasmic compartments at levels far below the K_m of CFAse. If this is correct, SAM levels, rather than lipid flux, could be the primary determinant of CFA formation when CFAse is outside mitochondria or the cytoplasm. This is a significant limitation on the general usefulness of METALIC. As it stands now, METALIC will certainly be useful for assessing lipid flux between mitochondria and the rest of the cell. For other compartments, using METALIC will require measurements of SAM levels in organelles, which is not trivial. One way around this is to localize CFAse on the cytoplasmic surface of organelles, but this will require determination of the stability of CFAse-containing fusions to rule out that they do not release free CFAse into the cytoplasm. These are not insurmountable problems, but they should be discussed more.

We agree the availability of SAM might be limiting in organelles other than mitochondria, nucleus and the cytoplasm. We acknowledge this clearly in the discussion now along with the mentioned reference. We however disagree that this limitation profoundly affects the usefulness of the approach. Indeed, of the 7 constructs we describe for CFAse expression, only one is potentially affected by this limitation, i.e. the ER-lumen targeted CFAse, all the other constructs targeting the cytosolic leaflet of the organelles (Fig 1c).

Our microscopy results suggest that the CFAse-containing fusions are stably targeted to organelles and any CFAse fraction released into cytoplasm is undetectable. We have added this point as well to the discussion. Finally, we also discuss the possibility to supply SAM in the medium so as to increase its availability in the endocytic compartment. Bulk endocytic flow allows the internalization of molecules as large as dextrans, and is the main way to label vacuoles (for instance using CMAC). There is no reason to believe that SAM wouldn't be internalized into early endosome/TGN, late endosomes and vacuoles. This is also something that we now discuss more in detail, if someone might need to target the luminal side of endocytic compartments.

2. I remain perplexed by the authors' interpretation of the findings in Fig. 5. The timing of formation of the double labeled PC species does not seem to support the claim that Mdm12 and Vps13 make significant contributions to lipid flux between mitochondria and the rest of the cell. If they did, then surely auxin treatment would cause a significant reduction in double labeled PC levels 3.5 hours after labeling started (the first time point). The fact that none is found cannot be explained by differences in growth rate, since the cells are still growing at the about the same rate when the 3.5-hour sample was taken (Fig. 4c). To me, this suggests none of the proteins tested makes a significant contribution. What happens at later times could be caused by factors other than changes in lipid flux, such as a decrease in SAM levels. All this is not to say that METALIC is not an interesting new method to study lipid flux, but

Fig. 5 is not a strong proof of concept.

We can actually exclude the possibility that a “decrease in SAM levels” accounts for the difference observed between wild-type and mutant strains. Indeed, when we assess the activity of either the SAM-dependent methyltransferase or SAM-dependent CFase alone (i.e. independently of each other, Fig. 5c, top two panels), the rates of labeling are virtually identical between the wt and the mutants, demonstrating that SAM levels do not decrease in the mutants at later time points.

Only the double labeling of PC, which requires ER-mitochondria transport, is affected in the mutants (Fig. 5c, bottom panel). These observations indicate that the observed differences are not due to a decrease in SAM levels, nor any metabolic change reflecting on the activity of either enzyme, but instead reflect the activity of ERMES/Vps13 in ER-mitochondria lipid transport..

Having said this, we do not fully understand what happens at 3.5h. We have a similar interpretation as Reviewer 3 that this must be due to some form of adaptation of yeast cells to the sudden exposure to a heavy dose of methionine to initiate labeling. We note additionally that even our negative control – Sam5-delete cells – display this bump in labelling at t3.5, before returning to background levels, indicating that this timepoint is most likely perturbed, rather than the subsequent ones.

In conclusion, analysis of singly-labeled species demonstrate the trustworthiness of the later time-points, while the analysis of Sam5-delete strains show the unreliability of the t3.5 timepoint. We have made this clearer in the text.

Reviewer #2 (Remarks to the Author):

The authors have satisfactorily dealt with all my concerns.

We thank the reviewer.

Reviewer #3 (Remarks to the Author):

The revised manuscript addresses the majority of concerns raised. The new Western blotting in Fig 2 makes it more clear how CFase organelle targeting influences enzyme abundance. The other major concerns have been addressed via comparisons to theoretical modeling of lipid turnover. Finally the addition of new data with CFase targeted to multiple mammalian organelles adds to the impact of the study.

We thank the reviewer.

COMMENTS ADDED DURING CROSS-COMMENTING, weighing in on Rev#1's points:

Regarding point 1, I agree that at present it makes sense to address this potential issue with more commentary in the discussion. The issue with SAM does potentially limit the general use of METALIC for certain organelles, and this limitation should be very clearly stated and discussed more.

This has been addressed now.

Regarding point 2...I see the concern here, however, my interpretation of the Fig 5C data was the 3.5hr plot was the yeast were still in their early growth phase of the experiment, and still coming out of their initial lag phase when transferred to the initial culture in the presence of auxin. Generally when inoculating yeast into fresh cultures for experiments like this, one may see the first few hours of the culture as in an "initial lag" phase as they prepare for logarithmic growth. This seems the case in Fig 4C, as their initial growth rate is slower at ~3.5hrs (all are equally slower, as noted by Rev1). At this time, initial membrane synthesis may be somewhat more supported from preexisting lipid stores like those in lipid droplets (and thus may not rely as extensively on inter-organelle transporters like Vps13). As growth rate picks up, this may change.

In any case, I think it may be something the authors should at least address by discussion. I struggle to think of better controls. I am curious what they will say regarding point 2, but as for me, I took that it had more to do with the initial yeast lag phase.

Kindly see response to Reviewer 1 (point 2)

Final Decision Letter:

Dear Dr Kornmann,

I am pleased to inform you that your manuscript, "METALIC reveals interorganelle lipid flux in live cells by enzymatic mass tagging", has now been accepted for publication in Nature Cell Biology. Congratulations on this interesting study!

28Over the next few weeks, your paper will be copyedited to ensure that it conforms to Nature Cell Biology style. Once your paper is typeset, you will receive an email with a link to choose the appropriate publishing options for your paper and our Author Services team will be in touch regarding any additional information that may be required.

Please note that *Nature Cell Biology* is a Transformative Journal (TJ). Authors may publish their research with us through the traditional subscription access route or make their paper immediately open access through payment of an article-processing charge (APC). Authors will not be required to make a final decision about access to their article until it has been accepted. Find out more about Transformative Journals

To assist our authors in disseminating their research to the broader community, our SharedIt initiative provides you with a unique shareable link that will allow anyone (with or without a subscription) to read the published article. Recipients of the link with a subscription will also be able to download and

29print the PDF.

If you have not already done so, we strongly recommend that you upload the step-by-step protocols used in this manuscript to the Protocol Exchange (www.nature.com/protocolexchange), an open online resource established by Nature Protocols that allows researchers to share their detailed experimental know-how. All uploaded protocols are made freely available, assigned DOIs for ease of citation and are fully searchable through nature.com. Protocols and Nature Portfolio journal papers in which they are used can be linked to one another, and this link is clearly and prominently visible in the online versions of both papers. Authors who performed the specific experiments can act as primary authors for the Protocol as they will be best placed to share the methodology details, but the Corresponding Author of the present research paper should be included as one of the authors. By uploading your Protocols to Protocol Exchange, you are enabling researchers to more readily reproduce or adapt the methodology you use, as well as increasing the visibility of your protocols and papers. You can also establish a dedicated page to collect your lab Protocols. Further information can be found at www.nature.com/protocolexchange/about

With kind regards,

Melina

Melina Casadio, PhD
Senior Editor, Nature Cell Biology
ORCID ID: <https://orcid.org/0000-0003-2389-2243>

** Visit the Springer Nature Editorial and Publishing website at www.springernature.com/editorial-and-publishing-jobs for more information about our career opportunities. If you have any questions please click here.**

30